# Gradient Inversion Transcript: Leveraging Robust Generated Priors to Reconstruct Training Data from Gradient Leakage

## Abstract

We propose Gradient Inversion Transcript (GIT), a novel model-based approach for reconstructing training data from leaked gradients. GIT employs a data reconstruction model, whose architecture is tailored to align with the inversion of the federated learning (FL) model's back-propagation process. Once trained offline, GIT can be deployed efficiently and only relies on the leaked gradients to reconstruct the input data, rendering it applicable under various distributed learning environments. When used as a prior for other iterative optimization-based methods, GIT not only accelerates convergence but also enhances the overall reconstruction quality. GIT consistently outperforms existing methods across multiple datasets and demonstrates strong robustness under challenging conditions, including inaccurate gradients, data distribution shifts and discrepancies in model parameters.

## 1 Introduction

In distributed learning (Jochems et al., 2016; McMahan et al., 2017; Yang et al., 2019) and federated learning (FL) (Huang et al., 2021), each client trains its model on local data and shares the gradients with a central server, which aggregates them to update the global model. While these methods are effective in improving performance and efficiency without directly exposing the client's data to public, recent research (Phong et al., 2017; Zhu et al., 2019; Zhao et al., 2020) has shown that the gradients leaked by sharing can still cause sensitive information leakage, as attackers may exploit them to reconstruct the original training data used by the individual client, posing significant privacy risks in real-world learning systems.

There is a considerable amount of works proposed to reconstruct the training data from its gradient (Huang et al., 2021; Phong et al., 2017; Zhu et al., 2019; Zhao et al., 2020; Wei et al., 2020; Geiping et al., 2020; Zhu & Blaschko, 2020; Wang et al., 2020; Yin et al., 2021; Wang et al., 2023; Jeon et al., 2021; Li et al., 2022; Fang et al., 2023; Wu et al., 2023; Chen & Vikalo, 2024; Wu et al., 2025) based on varying levels of model access, as shown in Table 1. These works generally fall into two major categories: *iterative optimization methods*, which iteratively optimize the reconstructed data to align its gradients with the leaked ones; and *model-based methods*, which leverage an auxiliary model to approximate the user data. Model-based methods can be further subdivided into two types: generative model-based methods and input-gradient mapping-based methods. The first type trains a latent space as the input of a pre-trained generative model. The second type trains a model mapping leaked gradients to the corresponding user data.

Iterative optimization methods typically require repeated access to gradients from the target model (Phong et al., 2017; Zhu et al., 2019; Wei et al., 2020; Geiping et al., 2020; Wang et al., 2020; Huang et al., 2021; Chen & Vikalo, 2024) or full access to the model parameters (Zhu & Blaschko, 2020; Wang et al., 2023). In contrast, the generative-model based methods (Yin et al., 2021; Jeon et al., 2021; Li et al., 2022; Fang et al., 2023; Wu et al., 2025) relies on a pre-trained generative model and public data that closely matches the distribution of the user data to train the latent space. And the input-gradient mapping-based methods (Wu et al., 2023) require input-gradient pairs from public datasets to train the auxiliary model itself.

We focus on the input-gradient mapping-based methods for input data reconstruction in this work. We call the model under attack the *leaked model*. The existing mapping-based method, such as

LTI (Wu et al., 2023), usually employs a fixed-architecture multi-layer perception (MLP) (Rosenblatt, 1958) as the data reconstruction model. However, this approach lacks justification for how gradients relate to the input data. We argue that the auxiliary model to reconstruct the input data from the gradient should approximate the inverse of the gradient computation process and thus be adaptive to the architecture of the leaked model. In this context, we introduce **gradient inversion transcript (GIT)** in this work to adaptively choose the architecture of the auxiliary reconstruction model to improve its effectiveness. In addition, we can combine GIT with iterative optimization methods like IG Geiping et al. (2020), in which we use GIT's output as the initial estimation of the input data for iterative optimization methods to further refine the reconstructed input data.

**Problem Settings:** In general, we assume that the attacker only has access to the leaked gradients and the model architecture. Our method does not need the parameters of the leaked model or the labels of the training data, as GIT trains the auxiliary reconstruction model on publicly available data with known labels. This direct input-gradient mapping architecture circumvents the need for dummy input optimization, thereby eliminating the requirement for label inference. As illustrated in Figure 1, we adopt a similar premise to DLG Zhu et al. (2019): the attacker hacks the channel to inject data to one client, which shares the gradient with the server and other clients. Alternatively, in scenarios where the attacker gains illicit access to the gradient query pipeline, our proposed GIT can also operate effectively in this context. The attack aims to reconstruct the data from *both the hacked client and other clients* by shared gradients. The problem settings and comparison with existing literature are summarized in Table 1.

**Practical Application Scenarios:** As shown in Table 1, all gradient leakage-based training data reconstruction methods require either gradient queries or access to known leaked model parameters. Our work adheres to the former assumption. This "curious-but-honest" setting is also commonly assumed in federated learning contexts, such as in data poisoning and backdoor attack scenarios.

**Challenges:** Figure 1 provides an overview of GIT. Beyond the basic case where the training data for auxiliary reconstruction model follows the same distribution as the input data for recovery, we may encounter more challenging scenarios: (1) The clients may send defensive inaccurate gradients, like clipped gradients Zhu et al. (2019) and noisy gradients Abadi et al. (2016); (2) When recovering data from other clients based on their shared gradients, challenges arise due to distributional shifts between data on different nodes and slight discrepancies in model parameters across nodes due to lack of synchronization. Input-gradient mapping-based methods can adapt to both scenarios, unlike iterative optimization-based methods,

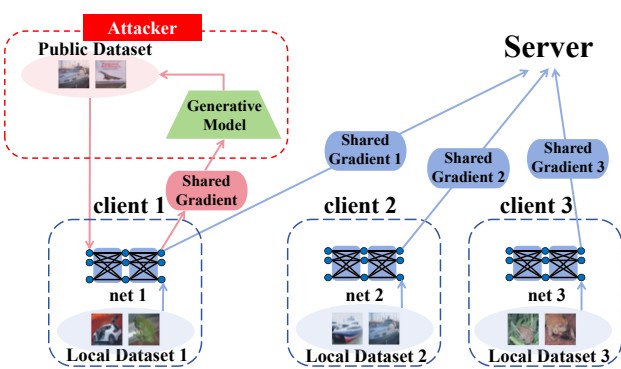

Figure 1: A flowchart of problem settings for GIT. The attacker hacks the channel of one client to inject data and utilizes the obtained input-gradient pair to train generative models. The attacker aims to reconstruct the data from *both the hacked client and other clients* by shared gradients.

which are not applicable for scenario (2) since the attacker has no access to inject data to unhacked clients and is therefore unable to reconstruct training data from them.

Compared with iterative optimization-based methods, our method can train reconstruction model offline and is much more efficient during deployment, making it well-suited for real-time reconstruction tasks with a small tolerated delay. Compared with generative–based methods, GIT demonstrates greater robustness to out-of-distribution (OOD) scenarios, as it focuses on inverting the backpropagation process and implicitly recovering the model parameters. Compared with other mapping-based methods like LTI (Wu et al., 2023), GIT is broadly applicable to leaked models with diverse architectures and achieves better performance. Moreover, GIT remains effective under challenging reconstruction scenarios.

We summarize our contributions as following points: (1) We propose *Gradient Inversion Transcript* **(GIT)**, which is inspired by back-propagation and constructs a reconstruction model whose architec-

ture is tailored to adapt the leaked model. GIT is shown to effectively reconstruct the input data given its gradient without the knowledge of model parameters and data labels. (2) GIT can be efficiently deployed after offline training. Compared with existing methods, GIT can achieve the best performance in most cases. In addition, the outputs of GIT can serve as the prior for gradient matching, further improving the performance. (3) GIT is generally applicable and has robust performance under some challenging performance. It remains effective under discrepancies in model parameters, and achieves best performance under inaccurate gradients and data distributional shift.

## 2 RELATED WORK

Table 1: The comparison in terms of attacker's access for different input data reconstruction methods. **dist.** means access to distribution of labels. The categories of methods are separated by dashed lines and, from top to bottom, are: parameter-based methods, iterative optimization-based methods, model-based methods (including generative model–based methods, and input-gradient mapping–based methods).

| Method | Model Parameters | Model Architecture | Shared Gradients | Gradient Query | Output Logit | Data Label | Public Data | Pretrained GAN |
|---|---|---|---|---|---|---|---|---|
| (RTNN) | ✓ | ✓ | ✗ | ✗ | ✗ | ✗ | ✗ | ✗ |
| (DLG; IG; Sapag; RLU) | ✗ | ✗ | ✓ | ✓ | ✗ | ✓ | ✗ | ✗ |
| (iDLG; iLRG) | ✗ | ✗ | ✓ | ✓ | ✗ | dist. | ✗ | ✗ |
| (Spear) | ✓ | ✓ | ✓ | ✗ | ✓ | ✓ | ✗ | ✗ |
| (R-gap; R-provably) | ✓ | ✓ | ✓ | ✗ | ✗ | ✓ | ✗ | ✗ |
| (GradInversion; DGGI) | ✗ | ✗ | ✓ | ✓ | ✗ | dist. | ✓ | ✓ |
| (GIAS; GGL; GIFD) | ✗ | ✗ | ✓ | ✓ | ✗ | ✓ | ✓ | ✓ |
| (LTI), GIT | ✗ | ✓ | ✓ | ✓ | ✗ | ✗ | ✓ | ✗ |

**Optimization-Based Methods.** The feasibility of optimization-based method for data reconstruction from gradient leakage was initially explored by Phong et al. (2017) Zhu et al. (2019) demonstrated its practicality by proposing Deep Leakage from Gradients (DLG). DLG optimizes a randomly generated dummy input to estimate the training data by matching its gradients and the leaked ground truth gradients. There are several subsequent works improving DLG from either optimization perspectives or more realistic scenarios Wei et al. (2020); Geiping et al. (2020); Wang et al. (2020); Zhu & Blaschko (2020); Wang et al. (2023); Chen & Vikalo (2024) with settings as shown in Table 1.

**Model-Based Methods.** Unlike optimization-based methods, model-based methods estimate the distribution of user data using an auxiliary model designed by the attacker, which maps the leaked gradient to input data estimation or the initial dummy input for subsequent optimization-based refinement. Model-based methods generally have two major categories, which are based on the generative model and the input-gradient mapping, respectively. The first type trains a latent space representation and uses a pre-trained generative model to synthesize estimations of the user data. The second type directly trains an auxiliary reconstruction model to map leaked gradients to the corresponding user data.

Early attempts of the generative method Yin et al. (2021) use a pre-trained generative model to produce image priors for reconstruction. Building on this, GIAS Jeon et al. (2021); Huang et al. (2021) employs a generative adversarial network (GAN) Goodfellow et al. (2014) as the generative model and alternately searches both the latent space and the parameter space of the generator. However, GIAS is computationally prohibitive, as it requires training a new generator for each reconstructed image. In this context, there are several works Li et al. (2022); Fang et al. (2023); Wu et al. (2025) focus on improving the efficiency and performance of GIAS under different settings as shown in Table 1.

The input-gradient mapping-based method was originally proposed by Wu et al. (2023) as Learning to Invert (LTI). Specifically, they design the reconstruction model as a three-layer multi-layer perceptron (MLP) with a fixed hidden size regardless of the leaked model, which may not be optimal. In contrast, we introduce gradient inversion transcript (GIT), which is a framework that dynamically selects the architecture of the threat model based on the leaked model to enhance performance.

## 3 INPUT DATA RECONSTRUCTION BY BACK-PROPAGATION

### 3.1 A GENERAL FRAMEWORK

As in Figure 2, we consider the generic architecture of a multi-input multi-output (MIMO) layer with nonlinear elementwise activation functions as follows:

$$\boldsymbol{z} = \sum_{i=1}^{N_{in}} \mathbf{W}_i^{(in)} \boldsymbol{a}_i^{(in)}, \ \boldsymbol{z}_j^{(out)} = \mathbf{W}_j^{(out)} \boldsymbol{a}, \tag{1}$$

$$\boldsymbol{a} = \sigma(\boldsymbol{z}), \ \boldsymbol{a}_j^{(out)} = \sigma_j(\boldsymbol{z}_j^{(out)}), \ j = 1, ..., N_{out}$$

Figure 2: An MIMO layer.

The MIMO layer is connected with $N_{in}$ input layers and $N_{out}$ output layers. $\sigma$ and $\{\sigma_i\}_{i=1}^{N_{out}}$ are the nonlinear activation functions. We let $B$ be the batch size, $\boldsymbol{z} \in \mathbb{R}^{B \times d}$ and $\boldsymbol{a} \in \mathbb{R}^{B \times d}$ represent the pre-activation and post-activation of this MIMO layer, respectively. Similarly, $\left\{ \left( \boldsymbol{z}_i^{(in)} \in \mathbb{R}^{B \times d_i^{(in)}}, \boldsymbol{a}_i^{(in)} \in \mathbb{R}^{B \times d_i^{(in)}} \right) \right\}_{i=1}^{N_{in}}$ and $\left\{ \left( \boldsymbol{z}_j^{(out)} \in \mathbb{R}^{B \times d_j^{(out)}}, \boldsymbol{a}_j^{(out)} \in \mathbb{R}^{B \times d_j^{(out)}} \right) \right\}_{j=1}^{N_{out}}$ represent the pre-activation and post-activation pairs for the input layers and the output layers, respectively. In addition, $\left\{ \mathbf{W}_i^{(in)} \in \mathbb{R}^{d \times d_i^{(in)}} \right\}_{i=1}^{N_{in}}$ and $\left\{ \mathbf{W}_j^{(out)} \in \mathbb{R}^{d_j^{(out)} \times d} \right\}_{j=1}^{N_{out}}$ refer to the weights connecting this layer and its adjacent layers. We replace notation $\mathbf{W}$ with $\boldsymbol{g}$ to represent the gradient of the loss function $\mathcal{L}$ w.r.t its weights, e.g., $\boldsymbol{g}_i^{(in)} = \nabla_{W_i^{(in)}} \mathcal{L}, \boldsymbol{g}_j^{(out)} = \nabla_{W_j^{(out)}} \mathcal{L}$. We omit the bias term for notation simplicity, since the bias terms can be incorporated as part of the weight matrices.

We have the following equations by back-propagation:

$$\boldsymbol{g}_j^{(out)} = \frac{\partial \mathcal{L}}{\partial \boldsymbol{z}_j^{(out)}} \otimes \boldsymbol{a}^T, \quad \boldsymbol{g}_i^{(in)} = \left( \left( \sum_{j=1}^{N_{out}} \mathbf{W}_j^{(out)T} \otimes \frac{\partial \mathcal{L}}{\partial \boldsymbol{z}_j^{(out)}} \right) \odot \sigma'(\boldsymbol{z}) \right) \otimes \boldsymbol{a}_i^{(in)T} \tag{2}$$

Here we define operator $\otimes$ as tensor multiplication and operator $\odot$ as broadcast row-wise product. In addition, $\boldsymbol{z}, \boldsymbol{a}$ are broadcast as a tensor of shape $B \times d \times 1$, similar broadcast mechanisms are applied to $\boldsymbol{z}_j^{(out)}$ and $\boldsymbol{z}_i^{(in)}$; $\mathbf{W}_j^{(out)}$ is broadcast as a tensor of shape $1 \times d_j^{(out)} \times d$, and the transpose operator $(\cdot)^T$ switches the second and the third dimensions of a 3-d tensor. Based on Equation (2), we cancel out $\partial \mathcal{L} / \partial \boldsymbol{z}_j^{(out)}$ and approximate the input of the layer as follows:

$$\boldsymbol{a}_i^{(in)T} \simeq \left( \left( \sum_{j=1}^{N_{out}} \mathbf{W}_j^{(out)T} \otimes \boldsymbol{g}_j^{(out)} \otimes (\boldsymbol{a}^T)^+ \right) \odot \sigma'(\boldsymbol{z}) \right)^+ \otimes \boldsymbol{g}_i^{(in)} \tag{3}$$

Here we use $(\cdot)^+$ to represent the Moore–Penrose inverse of a matrix. For a third-order tensor, $(\cdot)^+$ calculate the Moore-Penrose inverse of each of its subspace via the first dimension. Equation (3) establishes the formulation wherein we leverage the gradients, the parameters and the output activation to estimate the input data of a neuron. For a neural network of general architecture, we can estimate the input of each layer following back-propagation and ultimately obtain the reconstructed input data.

**Generality.** Our analysis is generic and can be applied to general neural network architectures as long as they support back-propagation. For multi-layer perceptrons (MLP) and vanilla convolutional neural networks (CNN) like LeNet, we have $N_{in} = N_{out} = 1$ for all layers; for residual networks (ResNet), we have $N_{out} > 1$ for layers which receive the inputs from both the preceding layer and the shortcut connections. Our framework is also compatible with more complicated architectures like attention mechanism in transformers Vaswani et al. (2017). We defer detailed derivation for these popular architectures in Appendix F.3. The mini-batch training setting is shown in Appendix F.1.

## 3.2 Modularized Input Data Reconstruction

For models with large amount of parameters, it would be computationally expensive to infer the input data by recursively using Equation (3). In this context, we formulate the large model as a composition of several modules and apply input data reconstruction on the module level instead of the layer level. We re-consider the multi-input multi-output (MIMO) layer as in Section 3.1 with input and output connections followed by functions $\left\{f_i^{(in)}\right\}_{i=1}^{N_{in}}$, $\left\{f_j^{(out)}\right\}_{j=1}^{N_{out}}$, respectively:

$$\boldsymbol{z} = \sum_{i=1}^{N_{in}} f_i^{(in)}(\mathbf{W}_i^{(in)}\boldsymbol{a}_i^{(in)}), \ \boldsymbol{z}_j^{(out)} = f_j^{(out)}(\mathbf{W}_j^{(out)}\boldsymbol{a}), \ j = 1, ..., N_{out}. \tag{4}$$

$\boldsymbol{a}_i^{(in)}$ and $\boldsymbol{a}$ are calculated in the same way as in Equation (1). We follow the derivation as in Section 3.1 and obtain the following equation for modularized input data reconstruction:

$$\boldsymbol{a}_i^{(in)T} = \left(\left(\sum_{j=1}^{N_{out}} \mathbf{W}_j^{(out)T} \otimes \boldsymbol{g}_j^{(out)} \otimes (\boldsymbol{a}^T)^+\right) \odot \sigma'(\boldsymbol{z}) \otimes f_i^{'(in)}(\mathbf{W}_i^{(in)}\boldsymbol{a}_i^{(in)})\right)^+ \otimes \boldsymbol{g}_i^{(in)} \tag{5}$$

Equation (5) demonstrates modularized input data reconstruction. It establishes a high-level formulation to estimate input data for large models. However, we need the gradient information from the input module $f_i^{(in)}$ to estimate $f_i^{'(in)}(\mathbf{W}_i^{(in)}\boldsymbol{a}_i^{(in)})$, which will be elaborated in the next section.

## 4 GIT: Gradient Inverse Transcript

**Exact-GIT.** Based on Equation (3) and the analyses in Section 3.1, the input value $\boldsymbol{a}_i^{(in)}$ of a general MIMO layer can be estimated from the activation $\boldsymbol{a}$, the gradient $\boldsymbol{g}_i^{(in)}$ of the input weight and output weights $\{\mathbf{W}_j^{(out)}\}_{j=1}^{N_{out}}$. Therefore, we can recursively utilize Equation (3) to reconstruct the input data by an auxiliary reconstruction model with all unknown variables, such as the weights, as its trainable parameters. We use the mean square error between the true input data and its estimation as the loss objective function. Once trained, the reconstruction model can subsequently reconstruct the training data batch using the leaked gradients as input during inference.

The detailed pseudo-code for the training and the inference phase is shown as Algorithm 1. The key innovation of our method is that we adaptively adjust the architectures of the reconstruction models by Equation (3) based on the leaked model and map the leaked gradients to the estimated input data, so we name it *gradient inverse transcript (GIT)*. We further name our method *Exact-GIT* when we strictly follow Equation (3), i.e. using model weights as parameters for the reconstruction models, for all layers to reconstruct the input data. We present some specific example architectures in Appendix F.3. The results of the Exact-GIT implementation are presented in Appendix F.2.

**Coarse-GIT.** Exact-GIT enjoys good interpretability but is computationally expensive for large models. Moreover, the Moore-Penrose inverse in Equation (3) would introduces numerical instability issues for large-scale tensors in practice. To tackle these issues, compared with Equation (3), we can also model such estimation in a more coarse-grained manner and name the corresponding method *Coarse-GIT*. Specifically, we utilize a shallow multi-layer perceptron (MLP) $m_\theta$, parameterized by $\theta$, to approximate the right-hand-side of Equation (3). The inputs of this shallow MLP are all the known variables on the right-hand-side of Equation (3), including the leaked gradients and the output activation. Therefore, like Equation (3), Coarse-GIT recursively estimates each layer's input by $\boldsymbol{a}_i^{(in)} = m_\theta\left(\{\boldsymbol{g}_j^{(out)}\}_{j=1}^{N_{out}}, \boldsymbol{g}_i^{(in)}, \boldsymbol{a}\right)$. The reconstruction model comprises multiple shallow MLPs, with orders based on back-propagation and collectively trained to minimize the difference between the estimated input and the corresponding ground truth.

Coarse-GIT also supports modularized input data reconstruction as discussed in Section 3.2, which is more computationally affordable. It employs a shallow MLP $m_\theta$ to estimate the right-hand-side of Equation (5): $\boldsymbol{a}_i^{(in)} = m_\theta\left(\{\boldsymbol{g}_j^{(out)}\}_{j=1}^{N_{out}}, \boldsymbol{g}_i^{f'(in)}, \boldsymbol{g}_i^{(in)}, \boldsymbol{a}\right)$ where $\boldsymbol{g}_i^{f'(in)}$ represents the leaked gradients for the parameters in the input module $f_i^{(in)}$. Compared with layerwise reconstruction,

---

Algorithm 1: Training and Inference of GIT

---

1: **Training input:** Training set of GIT, i.e., input-gradient pairs $\mathcal{D} = \left\{\left(\boldsymbol{x}^{(i)}, \boldsymbol{g}^{(i)}\right)\right\}_{i=1}^{N}$. Epoch budget $E$. Batch size $B$. Learning rate $\eta$.
2: Initialization: Construct the reconstruction model $M$ parameterized by $\Theta$ based on the architecture of the leaked model. Popular architectures are discussed as examples in Appendix F.3.
3: **for** the epoch index from 1 to $E$ **do**
4:    **for** the batch index from 1 to $N/B$ **do**
5:       Sample one mini-batch $\{\left(\boldsymbol{x}^{(b_i)}, \boldsymbol{g}^{(b_i)}\right)\}_{i=1}^{B}$
6:       **if** use *Exact-GIT* **then**
7:          Estimate the input $\{\widehat{\boldsymbol{x}}^{(b_i)}\}_{i=1}^{B} = M(\Theta, \{\boldsymbol{g}^{(b_i)}\}_{i=1}^{B})$ by recursively using Equation (3).
8:       **else**
9:          Estimate the input $\{\widehat{\boldsymbol{x}}^{(b_i)}\}_{i=1}^{B} = M(\Theta, \{\boldsymbol{g}^{(b_i)}\}_{i=1}^{B})$ by recursively using Equation (3) or Equation (5) with right hand side replaced by a shallow MLP discussed in Section 4.
10:       **end if**
11:       Calculate the loss $\mathcal{L}^{(gen)} = \frac{1}{2B}\sum_{i=1}^{B} \|\widehat{\boldsymbol{x}}^{b_i} - \boldsymbol{x}^{b_i}\|$ and update $\Theta \leftarrow \Theta - \eta\nabla_{\Theta}\mathcal{L}^{(gen)}$
12:    **end for**
13: **end for**
14: **Training output:** GIT generator $M$ with learned parameters $\Theta$.
15:
16: **Inference input:** GIT generator $M$ with parameters $\Theta$. Leaked gradients $\{\boldsymbol{g}^{(i)}\}_{i=1}^{N'}$.
17: **Inference output:** Input data estimation $\{\widehat{\boldsymbol{x}}^{(i)}\}_{i=1}^{N'} = M(\Theta, \{\boldsymbol{g}^{(i)}\}_{i=1}^{N'})$

---

modularized reconstruction only considers the high-level topologies of the leaked model, making it suitable for large models.

**Bootstrap.** For both Exact-GIT and Coarse-GIT, we need to estimate the output logits, i.e., last layer's output, to start the recursive estimation. The average output logits over the mini-batch can be analytically estimated if the last layer has a bias term Zhu & Blaschko (2020). Otherwise, we use the leaked gradients for the weight of the last layers to estimate the output logits by a shallow MLP. The ablation studies are presented in Appendix G.5.

## 5 EXPERIMENTS

We comprehensively assess our methods on various datasets, including CIFAR-10 (Krizhevsky et al., 2009), ImageNet (Deng et al., 2009) and facial datasets (Facial Expression Recognition (FER) from kaggle, Japanese Female Facial Expression (Jaffe) (Lyons et al., 1998)). Correspondingly, we employ various model architectures, including LeNet (LeCun et al., 1998), ResNet (He et al., 2016) and ViT (Dosovitskiy et al., 2020) to comprehensively demonstrate the effectiveness of our methods. Since the reconstruction models are trained by minimizing the mean square error (MSE) between the ground-truth and the estimated inputs, in addition to MSE, we also use peak signal-to-noise ratio (PSNR), structural similarity index (SSIM), learned perceptual image patch similarity (LPIPS) as metrics to quantitatively and comprehensively evaluate the performance of training data reconstruction. PSNR, SSIM and LPIPS reflect more perceptual and structural differences than MSE.

### 5.1 COMPARISON WITH BASELINES

We compare our method with baselines under two different settings: (1) we directly employ reconstruction models to map the leaked gradient to the reconstructed input data; (2) we first employ reconstruction models to obtain the input data estimation as priors and then refine the estimation by optimization-based methods. Unless specified, we use 10000 random samples and their gradients to train the reconstruction models. More implementation details are deferred to Appendix D.

#### 5.1.1 DIRECT INFERENCE BY AUXILIARY RECONSTRUCTION MODELS

As shown in Table 2, we first compare GIT with other reconstruction models in direct inference. Specifically, we compare GIT with Learning to Invert (LTI) Wu et al. (2023), which employs an MLP with approximately the same number of parameters as the generators. In addition, we include the

performance of optimization-based methods for reference, such as Deep Leakage from Gradients (DLG) Zhu et al. (2019) and Inverting Gradients (IG) Geiping et al. (2020). The computational overhead for optimization-based methods and mapping-based methods are fundamentally different. Optimization-based methods necessitate a complete optimization process for each batch data recovery, whereas input-gradient mapping-based methods need to train an auxiliary reconstruction model capable of retrieving data from the corresponding leaked gradients. We run both types of methods until their convergence and report the training and inference time for comparison.

The results in Table 2 include different tasks and network architectures. To save memory consumption and guarantee numerical stability, we adopt Coarse-GIT for all architectures and specifically modularized reconstruction for ViT. The GIT implementation details for specific architectures are deferred to Appendix D. The results indicate that GIT outperforms in most cases and metrics than baselines, including both optimization-based methods and mapping-based methods.

Visual inspection of reconstructed ImageNet samples by GIT as shown in Appendix G.7 reveals that images with large uniform color regions tend to be recovered more accurately, while those containing complex structures or multiple objects exhibit inferior reconstruction quality. This is consistent with the observations in Table 2 that GIT always performs the best in term of MSE but may underperform in term of LPIPS which focuses more on the image structure. Therefore, instead of directly employing GIT for inference, we further utilize it as an image prior to guide optimization-based methods toward more perceptually accurate results.

Table 2: Quantitative comparison for different datasets and models in terms of different metrics. Dashed lines separate mapping-based methods with optimization-based ones. The training time represents the time cost for training the reconstruction model. The inference time represents the average time to reconstruct one input data instances from the leakage gradients during inference.

| Dataset | Leaked Model | Method | MSE↓ | PSNR↑ | LPIPS↓ | SSIM↑ | Training Time (s) | Inference Time (s) |
|---|---|---|---|---|---|---|---|---|
| CIFAR10 | LeNet | DLG | 0.073 | 11.32 | 0.2380 | 0.0847 | / | 1660 |
| | | IG | 0.082 | 11.27 | 0.3916 | 0.1193 | / | 1899 |
| | | LTI | 0.015 | 19.17 | **0.2202** | 0.5304 | 8549 | 0.0030 |
| | | GIT | **0.010** | **20.38** | 0.2663 | **0.5533** | 8071 | 0.0025 |
| | ResNet | DLG | 0.084 | 10.93 | 0.3813 | 0.0667 | / | 7474 |
| | | IG | 0.080 | 10.75 | **0.2489** | 0.0739 | / | 6875 |
| | | LTI | 0.035 | 15.32 | 0.4400 | 0.2888 | 5212 | 0.0020 |
| | | GIT | **0.032** | **15.53** | 0.3957 | **0.3188** | 4019 | 0.0013 |
| ImageNet | ResNet | DLG | 0.147 | 9.25 | 0.8754 | 0.1324 | / | 3974 |
| | | IG | 0.161 | 9.17 | 0.8802 | 0.1283 | / | 4103 |
| | | LTI | 0.043 | 14.25 | 0.9017 | 0.3418 | 10200 | 0.0007 |
| | | GIT | **0.039** | **14.42** | **0.8513** | **0.3507** | 13011 | 0.0008 |
| | ViT | DLG | 0.172 | 7.57 | 0.9513 | 0.1217 | / | 3734 |
| | | IG | 0.175 | 7.64 | 0.9427 | 0.1210 | / | 3025 |
| | | LTI | 0.046 | 13.37 | 0.9223 | 0.2117 | 9738 | 0.0029 |
| | | GIT | **0.034** | **15.25** | **0.8365** | **0.3774** | 6717 | 0.0025 |

### 5.1.2 OPTIMIZATION-BASED DATA RECONSTRUCTION USING GIT AS PRIORS

Optimization-based methods are shown highly sensitive to the initialization of dummy inputs Wei et al. (2020). Therefore, recent methods, such as Gradient Inversion with Generative Image Prior (GIAS) Jeon et al. (2021), propose to utilize generative models to generate image priors as initialization of optimization-based methods. In this context, we can employ GIT to generate informative priors, which are subsequently refined through iterative optimization-based methods like IG. For model-based methods (Yin et al., 2021; Jeon et al., 2021; Li et al., 2022; Fang et al., 2023; Wu et al., 2025; 2023), we select GIAS Jeon et al. (2021) and LTI Wu et al. (2023) as a representative baseline for comparison. To ensure fairness, all reconstruction models are trained from scratch without relying on any pretraining, and are followed by the same optimization-based method.

As shown in Table 3, using GIT to generate priors and refine the reconstructed image by IG (GIT+IG) have the best performance in almost all cases and all metrics. In addition, GIT+IG always has better performance than LTI+IG, indicating GIT based on adaptive architectures can provide better

priors than LTI based on fixed architectures. Furthermore, GIT+IG, as a hybrid approach, not only converges faster but also outperforms both GIT and IG when used individually, demonstrating its superior effectiveness. We visualize the convergence curves of IG with and without the image prior in Appendix G.6, further highlighting the benefits of incorporating generated priors into the optimization process.

Table 3: Quantitative comparison for different datasets and models in terms of different metrics. The performance of IG is used as references. The training time represents the time cost for training the reconstruction model. The inference time represents the average time to reconstruct one input data instances from the leakage gradients during inference.

| Dataset | Leaked Model | Method | MSE↓ | PSNR↑ | LPIPS↓ | SSIM↑ | Training Time (s) | Inference Time (s) |
|---------|--------------|--------|------|-------|--------|-------|-------------------|--------------------|
| CIFAR10 | LeNet | IG | 0.082 | 11.27 | 0.3916 | 0.1193 | / | 1899 |
| | | GIAS+IG | 0.009 | 21.45 | 0.0328 | 0.8925 | 10025 | 242 |
| | | LTI+IG | 0.002 | 30.86 | 0.0025 | 09356 | 8549 | 158 |
| | | GIT+IG | **0.001** | **31.25** | **0.0009** | **0.9551** | 8071 | 161 |
| | ResNet | IG | 0.080 | 10.75 | 0.2489 | 0.0739 | / | 6875 |
| | | GIAS+IG | 0.019 | 18.96 | 0.2437 | 0.6125 | 10892 | 187 |
| | | LTI+IG | 0.009 | 20.48 | 0.0092 | 0.8266 | 5212 | 1651 |
| | | GIT+IG | **0.002** | **31.34** | **0.0041** | **0.9218** | 4019 | 1655 |
| ImageNet | ResNet | IG | 0.161 | 9.17 | 0.8802 | 0.1283 | / | 4103 |
| | | GIAS+IG | 0.037 | 14.32 | 0.8218 | 0.3765 | 27453 | 3209 |
| | | LTI+IG | 0.029 | 15.38 | 0.7434 | 0.4129 | 10200 | 2065 |
| | | GIT+IG | **0.021** | **16.78** | **0.6995** | **0.4758** | 13011 | 1998 |
| | ViT | IG | 0.175 | 7.64 | 0.9427 | 0.1210 | / | 3025 |
| | | GIAS+IG | 0.039 | 14.09 | 0.7572 | **0.5239** | 36950 | 3997 |
| | | LTI+IG | 0.029 | 15.96 | 0.7250 | 0.4231 | 6138 | 2950 |
| | | GIT+IG | **0.019** | **17.21** | **0.6730** | 0.5025 | 6717 | 2987 |

## 5.2 RECONSTRUCTION UNDER CHALLENGING SITUATIONS

In this section, we investigate the robustness of reconstruction methods under different challenging situations, including inaccurate leaked gradients and the substantial distributional shift between the public data and the training data. In such situations, optimization-based methods are not applicable or do not have competitive performance. Therefore, we mainly compare the results from the direct inference by model-based methods. More implementation details are deferred to Appendix D.

### 5.2.1 INACCURATE GRADIENTS

Prior works Wu et al. (2023) have shown degraded performance of optimization-based methods when the leaked gradients are inaccurate. Unlike optimization-based methods which observe significant performance degradation in the case of inaccurate gradients, Appendix G.2 shows that input-gradient mapping-based methods primarily utilize the gradient elements with large absolute values to generate outputs, indicating robustness in such challenging cases.

In Table 4, we consider the leaked gradients perturbed by isotropic Gaussian noise with different standard deviation (std). I compare different input-

Table 4: Comparison of metrics under gradient perturbation with varying noise variance. The batch size is fixed at 1, and the leaked model is LeNet with 5 layers.

| Method | std of noise | MSE↓ | PSNR↑ | LPIPS↓ | SSIM↑ |
|--------|--------------|------|-------|--------|-------|
| IG | None | 0.082 | 11.27 | 0.3916 | 0.1193 |
| | 0.01 | 0.105 | 9.79 | 0.4098 | 0.1172 |
| | 0.1 | 0.162 | 9.18 | 0.4320 | 0.1126 |
| LTI | None | 0.015 | 19.17 | 0.2202 | 0.5304 |
| | 0.01 | 0.015 | 19.16 | 0.2205 | 0.5300 |
| | 0.1 | 0.015 | 19.16 | 0.2199 | 0.5287 |
| GIAS | None | 0.012 | 19.21 | 0.2350 | 0.5398 |
| | 0.01 | 0.012 | 19.18 | 0.3010 | 0.5219 |
| | 0.1 | 0.013 | 18.87 | 0.3113 | 0.5189 |
| GIT | None | 0.010 | 20.38 | 0.2663 | 0.5533 |
| | 0.01 | 0.010 | 20.36 | 0.2675 | 0.5520 |
| | 0.1 | 0.010 | 20.37 | 0.2669 | 0.5522 |

gradient mapping-based methods and also include the performance of optimization-based methods like IG for reference. The results confirm the vulnerability of optimization-based methods against

gradient perturbations. Among input-gradient mapping-based methods, GIT performs the best in all cases and all metrics, showing minimal susceptibility to inaccurate gradients.

### 5.2.2 DISTRIBUTION SHIFT

As shown in Figure 1, GIT is trained on the public dataset injected by the attacker, and aims to reconstruct the local dataset. There may be a distributional shift between these two datasets, which could influence the effectiveness of the reconstruction model.

We consider two possible scenarios of distribution shifts: (1) the public and local datasets come from different subsets of the same dataset, with distribution differences arising from overlapping but distinct classes; (2) the public datasets are subsets of huge but more general datasets, such as FER, while the local datasets held by individual clients are more specific ones, such as Jaffe.

Our experiment in Table 5 investigate both scenarios above. For CIFAR-10 and ImageNet, the public data and the local data share 6 classes and the rest classes are distinct. For facial dataset, the public data and the local data have different resolutions and significant distributional shifts. The results in Table 5 indicate that GIT demonstrates the strongest generalization ability across distribution shifts and achieves the best performance on the local dataset. GIT learns an implicit representation of the leaked model's parameters by its adaptive architecture, which is more agnostic to the data distribution. By contrast, GIAS learns a latent space that captures the distribution characteristics of the public dataset, which requires the public and local datasets to share highly similar features to perform well.

Table 5: Comparison of the metrics under distributional shift. For each dataset, we select a federated learning model architecture that is well-suited for its classification task: LeNet is used for CIFAR-10, ResNet for ImageNet, and Vision Transformer (ViT) for facial datasets. The "classes" in the public data and local data represent categories sampled to form datasets. In the case of facial data, we conduct experiment where both the public data and local data come from FER, which serves as a comparison.

| Dataset | Public Data | Local Data | Method | MSE↓ | PSNR↑ | LPIPS↓ | SSIM↑ |
|---------|-------------|------------|--------|------|-------|--------|-------|
| CIFAR10 | classes 1-8 | classes 3-10 | GIAS | 0.065 | 11.87 | 0.3670 | 0.3092 |
| | | | LTI | 0.029 | 15.38 | 0.3028 | 0.3790 |
| | | | GIT | **0.020** | **17.44** | **0.2155** | **0.4150** |
| ImageNet100 | classes 1-53 | classes 48-100 | GIAS | 0.061 | 12.15 | 0.9518 | 0.3126 |
| | | | LTI | 0.049 | 13.10 | 0.9274 | 0.3150 |
| | | | GIT | **0.043** | **13.67** | **0.9044** | **0.3224** |
| Facial Data | FER | FER | GIAS | 0.020 | 17.73 | 0.4174 | 0.4051 |
| | | | LTI | 0.020 | 17.60 | 0.4420 | 0.3949 |
| | | | GIT | **0.018** | **17.93** | **0.3405** | **0.4228** |
| | | Jaffe | GIAS | 0.042 | 13.77 | 0.5128 | 0.2826 |
| | | | LTI | 0.033 | 15.21 | 0.4625 | 0.3187 |
| | | | GIT | **0.030** | **15.54** | **0.3461** | **0.3244** |

### 5.3 MORE ANALYSES AND ABLATION STUDIES

More analyses and ablation studies are deferred to Appendix G.

## 6 CONCLUSIONS

This work introduces *Generative Gradient Inversion Transcript (GIT)*, a novel method for reconstructing training data in federated learning by exploiting gradient leakage. We propose a reconstruction framework with an adaptive structure inspired by the inverse of backpropagation. GIT offers a significant efficiency advantage, being more cost-effective in inference than optimization-based methods, and can be seamlessly deployed after offline training. Compared to existing methods, GIT achieves superior performance in most cases. Furthermore, GIT-generated outputs can serve as priors for optimization-based gradient matching approaches, further enhancing attack effectiveness. GIT demonstrates strong robustness under challenging conditions, including inaccurate gradients, distributional shifts, and discrepancies in model parameters across clients. Future work will focus on extending GIT into a more generalized autoencoder framework and enhancing its reconstruction capabilities.

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

## A    ETHICS STATEMENT

This work proposes a novel method, *Generative Gradient Inversion Transcript (GIT)*, for reconstructing training data from gradients in federated learning. The study is conducted with the primary goal of exposing and understanding potential privacy risks associated with gradient leakage, a critical challenge in distributed learning paradigms. All experiments are performed using standard models and architectures on publicly available datasets; no real-world sensitive or private data is involved. We strongly discourage any malicious use of this method and advocate for its responsible application in security and privacy research.

## B    REPRODUCIBILITY STATEMENT

To ensure the reproducibility of our work, we will provide anonymized supplementary materials. These will include the complete implementation code of the GIT framework. The configuration details are shown in Appendix D.

## C    THE USE OF LARGE LANGUAGE MODELS (LLMs)

In the preparation of this manuscript, large language models (LLMs) were utilized solely as auxiliary tools for improving writing quality. Their use was strictly limited to polishing the text, such as correcting grammatical errors, enhancing sentence clarity, and ensuring consistent formatting. No LLM was involved in generating the core research ideas, designing the GIT methodology, conducting analyses, or drawing conclusions. All text that received LLM-assisted editing was meticulously reviewed and substantively refined by the authors, who take full responsibility for the entire intellectual content of this paper.

## D    EXPERIMENT CONFIGURATIONS

**Universal Settings** We employ various architectures for the leaked model. For LeNet, we use a 5-layer configuration with kernel size 2 and same padding. For ResNet, we adopt a 15-layer variant with kernel size 3, consisting of 4 blocks, each containing 2 convolutional layers and 1 skip connection. For ViT, we connect four 4-head attention blocks following the patch embedding layer.

For generative methods, we use 10000 batches of input-gradient pairs from the public dataset to train the generative model. During reconstruction, we use 10000 batches of gradients from the local dataset to recover the corresponding local data. For iterative optimization-based methods, we perform reconstruction for each batch of local data by starting from dummy inputs and applying iterative optimization individually.

For Coarse-GIT and Module-GIT, we use $m_\theta$ and $f_\vartheta$ with 3000 neurons in each hidden layer. For LTI, we employ a generative model consisting of three hidden layers, each with 3000 neurons, as described in Wu et al. (2023).

**Specific Settings** In our experiments described in Section 5.1.1, the inference time for generative methods is computed as the average over reconstructing 10000 local data batches, whereas for optimization-based methods, it is calculated based on the average over 10 local data batches, since each reconstruction is significantly more time-consuming.

In our experiments described in Section 5.1.2, the inference time of generative+optimization-based hybrid methods is computed as the sum of their individual inference times. For all methods, inference time is calculated as the average over 10 local data batches.

# E  NOTATION

| | |
|---|---|
| $\mathcal{L}$ | Loss objective function of the leaked model |
| $\mathcal{L}^{(gen)}$ | Loss objective function of GIT |
| $\sigma(\cdot)$ | The elementwise activation function |
| $\sigma'(\cdot)$ | The derivative of $\sigma(\cdot)$ |
| $\boldsymbol{z}$ | Pre-activation of an MIMO layer |
| $\boldsymbol{a}$ | Post-activation of an MIMO layer |
| $\mathbf{W}$ | Weight matrix in a neural network |
| $\boldsymbol{g}$ | Gradient of the loss objective function w.r.t $\mathbf{W}$ |
| $B$ | Batch size in mini-batch training |
| $b$ | Batch index in mini-batch training |
| $\mathbb{E}$ | Empirical average over the samples in batch $b$ |
| $d$ | Number of hidden nodes of an MIMO layer |
| $N_{in}$ | Number of input layers of an MIMO layer |
| $N_{out}$ | Number of output layers of an MIMO layer |
| $N$ | Number of input-gradient pairs for training GIT |
| $N'$ | Number of input-gradient pairs for testing |
| $(\cdot)^{+}$ | Moore-Penrose inverse of a matrix; or Moore-Penrose of each of $(\cdot)$'s subspace via the first dimension when $(\cdot)$ is a third order tensor |
| $(\cdot)^{T}$ | Transpose of a matrix; or transpose of the second and the third dimension when $(\cdot)$ is a third order tensor |
| $\otimes$ | Tensor Multiplification |
| $\odot$ | Broadcast row-wise product |
| $f(\cdot)^{(in)}$ | A module that approximates the input mapping of an MIMO layer |
| $f(\cdot)^{(out)}$ | A module that approximates the output mapping of an MIMO layer |
| $f'(\cdot)$ | The derivative of module $f(\cdot)$ |
| $\mathcal{D}$ | Input-gradient pairs |
| $E$ | Epoch budget |
| $\eta$ | Learning rate |
| $(\cdot)^{(i)}$ | The $i$-th sample in the dataset |
| $M$ | The generative model GIT |
| $\Theta$ | Trainable parameters of GIT |
| $m_{\theta}$ | A shallow MLP parameterized by $\theta$ to approximate recursive reconstruction in Coarse-GIT |
| $\vartheta$ | Trainable parameters in Module-GIT |
| $\boldsymbol{x}$ | Input data of the leaked model |
| $\hat{\boldsymbol{x}}$ | The estimated input data by GIT |

# F  METHODOLOGY DETAILS

## F.1  MINI-BATCH TRAINING.

In mini-batch training, $\boldsymbol{g}_i^{(in)}$ and $\boldsymbol{g}_j^{(out)}$ obtained by Equation (2) contains the gradient information for all data instances in the mini-batch. In practice, the leaked the gradient is their average over the mini-batch, that is $\boldsymbol{g}_i^{(in)} \leftarrow \mathbb{E}_b \boldsymbol{g}_i^{(in)}[b,:,:]$, $\boldsymbol{g}_j^{(out)} \leftarrow \mathbb{E}_b \boldsymbol{g}_j^{(out)}[b,:,:]$. Since the leaked gradient is the average over the mini-batch. When reconstructing the input data, we broadcast the leaked gradient in the dimension of batch size in Equation (3).

## F.2  EXACT-GIT

**Activation Function.** The Exact-GIT method in Algorithm 1 requires iteratively applying Equation (3). Equation (3) involves the derivative of the activation function $\sigma'(\boldsymbol{z})$, which can be estimated by $\boldsymbol{a}$. Although function $\sigma$ may not be an injective function, we demonstrate in Table 6 below that we can uniquely identify $\sigma'(\boldsymbol{z})$ given $\boldsymbol{a}$ for the most popular activation functions used in practice. In

Table 6: Mappings from $\boldsymbol{a}$ to $\sigma_i'(\boldsymbol{z})$ for popular activation functions. Operations are elementwise.

| Name | ReLU | Leaky ReLU | Sigmoid | Tanh |
|---|---|---|---|---|
| $\boldsymbol{a} = \sigma(\boldsymbol{z})$ | $\max(0, \boldsymbol{z})$ | $\max(k\boldsymbol{z}, \boldsymbol{z})$ | $\frac{1}{1+e^{-\boldsymbol{z}}}$ | $\frac{e^{\boldsymbol{z}} - e^{-\boldsymbol{z}}}{e^{\boldsymbol{z}_i} + e^{-\boldsymbol{z}}}$ |
| $\sigma'(\boldsymbol{z})$ | $\begin{cases} 1 & \text{if } \boldsymbol{a} > 0 \\ 0 & \text{if } \boldsymbol{a} = 0 \end{cases}$ | $\begin{cases} 1 & \text{if } \boldsymbol{a} > 0 \\ k & \text{if } \boldsymbol{a} \leq 0 \end{cases}$ | $\boldsymbol{a}_i(1 - \boldsymbol{a})$ | $1 - \boldsymbol{a}^2$ |

Exact-GIT, the weights of the generative attack model represent the estimated weights of the leaked model. Therefore, we can compare the difference between their weights to investigate to which degree the generative attack models recover the gradient-to-input inversion. In this context, we run Exact-GIT based on Algorithm 1 and plot its convergence curve as in Figure 3. Figure 3 illustrates the $l_2$ distance curve between the generative model's weights and the leaked model's weights, alongside the MSE between the reconstructed inputs and the ground truth inputs. As shown in Figure 3, when Exact-GIT converges, its weights align closely with the ground truth weights. This convergence highlights the effectiveness of exact-GIT in extracting weight information from the leaked model.

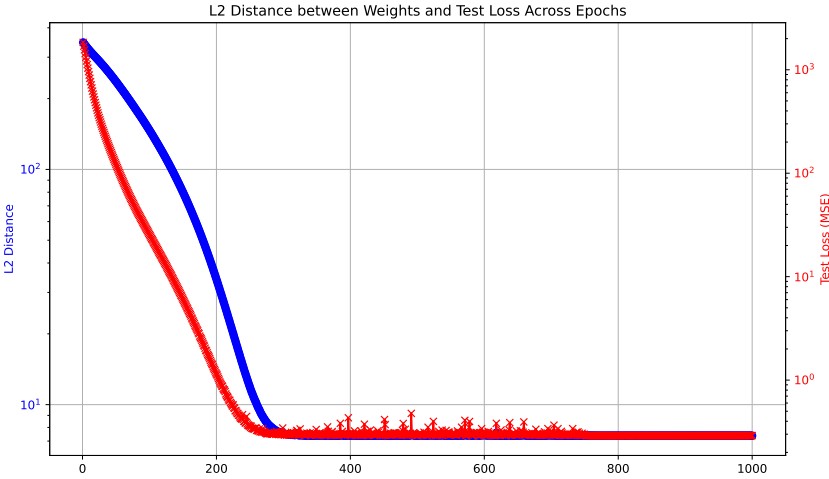

Figure 3: The red curve represents convergence curve of $l_2$ distance between weights of the generative model and the leaked model. The blue curve represents the convergence curve of MSE between reconstructed input and the ground truth input. The experiment is conducted on CIFAR-10 using Exact-GIT.

### F.3 COARSE-GIT FOR DIFFERENT ARCHITECTURES

Section 3 demonstrates a generic framework for any neural network architectures as long as they support back-propagation. In this section, we provide more technical details for specific neural network architecture that we use in the experiments, including feed forward networks, residual networks (ResNet) and vision transformer (ViT). We believe the details in this section will provide more insights for practitioners to understand how GIT is adapted to different neural architectures. Due to the scale of the architectures discussed in this section, we employ Coarse-GIT for all of them.

While related to the notation we use in Section 3, we use specific notations in this section for better readability. We provide the exact definition for each of these notations.

#### F.3.1 FEED FORWARD NEURAL NETWORKS

Feed forward neural networks, including multi-layer perceptron (MLP) and convolutional neural networks (CNN), can be formulated as follows:

$$
\begin{aligned}
\mathcal{L}_\theta(\boldsymbol{x}, y) &= \ell(\boldsymbol{z}_N, y) = \ell(\mathbf{W}_N \boldsymbol{a}_{N-1}, y) \\
\boldsymbol{a}_i &= \sigma_i(\boldsymbol{z}_i), \ \boldsymbol{z}_i = \mathbf{W}_i \boldsymbol{a}_{i-1}, \ i = 1, 2, ..., N-1
\end{aligned}
\tag{6}
$$

We denote the number of hidden nodes for the $i$-th layer as $\{d_i\}_{i=1}^{N-1}$. The input data batch $\boldsymbol{a}_0 = \boldsymbol{x} \in \mathbb{R}^{B \times d_0}$, where $B$ is the batch size. $\theta = \{\mathbf{W}_i \in \mathbb{R}^{d_i \times d_{i-1}}\}_{i=1}^N$ refer to the parameters of $N$ linear layers, including convolutional layers and fully connected layers. $\{\sigma_i\}_{i=1}^{N-1}$ are the nonlinear activation functions of different layers. In this context, $\{\boldsymbol{z}_i \in \mathbb{R}^{B \times d_i}\}_{i=1}^{N-1}$ and $\{\boldsymbol{a}_i \in \mathbb{R}^{B \times d_i}\}_{i=1}^{N-1}$ represent the pre-activation and post-activation of intermediate layers, respectively. $\boldsymbol{z}_N = \mathbf{W}_N \boldsymbol{a}_{N-1}$ is the output logit, and $\ell$ is the function calculating the classification error, such as the softmax cross-entropy function. We use $\boldsymbol{g}_i = \nabla_{\mathbf{W}_i} \mathcal{L}_\theta(\boldsymbol{x}, y)$ to represent the gradient of each weight matrix.

In this context, similar to Equation (2) and Equation (3) in Section 3, we derive the back-propagation and then the iterative input layer approximation for feed forward neural network defined in Equation (6) as follows:

$$
\boldsymbol{g}_i = \prod_{j=i}^{N-1} \left( \mathbf{W}_{j+1}^T \odot \sigma_j'(\boldsymbol{z}_j) \right) \otimes \frac{\partial \mathcal{L}}{\partial \boldsymbol{z}_N} \otimes \boldsymbol{a}_{i-1}^T
\tag{7}
$$

$$
\boldsymbol{a}_{i-1}^T \simeq \boldsymbol{a}_i^T \otimes \boldsymbol{g}_{i+1}^+ \otimes (\mathbf{W}_{i+1}^T \odot \sigma_i'(\boldsymbol{z}_i))^+ \otimes \boldsymbol{g}_i
\tag{8}
$$

$\otimes$ and $\odot$ have the same definition as in Section 3. As we can see, Equation (7) can be considered as a specific case of Equation (3) where $N_{in} = N_{out} = 1$ Furthermore, we employ Coarse-GIT in the experiments. Specifically, we use an MLP model $f$ parameterized by $\vartheta$ to estimate $\boldsymbol{a}_{i-1}$ from $\boldsymbol{a}_i$, $\boldsymbol{g}_{i+1}$ and $\boldsymbol{g}_i$. We apply Equation (7) recursively and utilize it to reconstruct the input data.

$$
\boldsymbol{a}_{i-1} = f_\vartheta(\boldsymbol{a}_i, \boldsymbol{g}_{i+1}, \boldsymbol{g}_i)
\tag{9}
$$

#### F.3.2 RESIDUAL NETWORKS

The key feature for residual networks (ResNet) He et al. (2016) is the skip connections, resulting in $N_{out} > 1$ for layers that combine inputs from both the previous layer and shortcut connections.

Without the loss of generality, we generally follow the notation of feed forward neural network defined in (6) except that there is a single shortcut connection linking the $k$-th layer to $l$-th layer ($k < l$). Specifically, the shortcut connection links the post-activation $\boldsymbol{a}_k$ to the pre-activation $\boldsymbol{z}_l$ with a weight parameter $\mathbf{S} \in \mathbb{R}^{d_k \times d_l}$. Therefore, $\{\boldsymbol{z}_i\}_{i=1}^N$ and $\{\boldsymbol{a}_i\}_{i=1}^N$ are calculated in the same manner except that $\boldsymbol{z}_l = \mathbf{W}_l \boldsymbol{a}_{l-1} + \mathbf{S} \boldsymbol{a}_k$. Based on the back propagation, $\boldsymbol{g}_i$ is calculated in the same way as in Equation (7) when $i > k$. When $i \leq k$, $\boldsymbol{g}_i$ is calculated as follows:

$$
\boldsymbol{g}_i = \prod_{j=i}^{k-1} M_j \otimes \left( \prod_{j=k}^{l-1} M_j + \mathbf{S} \odot \sigma_k'(\boldsymbol{z}_k) \right) \otimes \prod_{j=l}^{N-1} \left( \mathbf{W}_{j+1}^T \odot \sigma_j'(\boldsymbol{z}_j) \right) \otimes \frac{\partial \mathcal{L}}{\partial \boldsymbol{z}_N} \otimes \boldsymbol{a}_{i-1}^T
\tag{10}
$$

Following a similar analysis to feed forward neural networks, we can derive an approximation of $\boldsymbol{a}_{i-1}$ using $\boldsymbol{a}_i$. The approximation is the same as (8) except for the case $i = k$. This is because we calculate $\boldsymbol{a}_i$ using $\boldsymbol{a}_{i-1}$ in the same manner except for the case $i = k$, where the $k$-th layer is

connected to not only the immediate preceding layer but also the $l$-th layer via shortcut connection. Therefore, $\boldsymbol{a}_{k-1}$ is approximated in a different way from (8) as follows.

$$\boldsymbol{a}_{k-1} \simeq \left(\mathbf{W}_{k+1}^T \odot \sigma_k'(\boldsymbol{z}_k)\right) \otimes \boldsymbol{g}_{k+1} \otimes (\boldsymbol{a}_k^T)^+ + (\mathbf{S} \odot \sigma_k'(\boldsymbol{z}_k)) \otimes \boldsymbol{g}_l \otimes (\boldsymbol{a}_{l-1})^+ \otimes \boldsymbol{g}_k \quad (11)$$

Compared with (8), the estimation in (11) incorporates not only $\boldsymbol{g}_k$ and $\boldsymbol{g}_{k+1}$ but also $\boldsymbol{g}_l$ to estimate $\boldsymbol{a}_{k-1}^T$, which is consistent with the case of $N_{out} = 2$ in the analysis in Section 3. Since $\boldsymbol{a}_k$ is connected to $\boldsymbol{z}_l$ via skip connection, gradients can flow directly from the $k$-th layer to the $l$-th layer in back propagation. The insight provided by the approximation in Equation (11) indicates that the reconstruction sequence follows the same path as the gradient flow during backpropagation.

We use Coarse-GIT in the experiment for ResNet, similar to Equation (9), we employ an MLP model $f$ parameterized by $\vartheta$ and reconstruct $\boldsymbol{a}_{k-1}$ by:

$$\boldsymbol{a}_{k-1} = f_\vartheta(\boldsymbol{a}_k, \boldsymbol{g}_{k+1}, \boldsymbol{g}_k, \boldsymbol{g}_l) \quad (12)$$

When estimating the input from the leaked gradients, we apply Equation (12) when there is a shortcut connection and Equation (9) otherwise.

### F.3.3 VISION TRANSFORMER (VIT)

In the case of vision transformer (ViT) (Dosovitskiy et al., 2020), we apply modularized input data reconstruction and represent the each self-attention module as follows:

$$\boldsymbol{z} = \text{softmax}\left(\frac{\mathbf{Q}\mathbf{K}^\top}{\sqrt{d_k}}\right)\mathbf{V}, \quad \mathbf{Q} = \boldsymbol{a}_i^{(in)}\mathbf{W}^Q, \quad \mathbf{K} = \boldsymbol{a}_i^{(in)}\mathbf{W}^K, \quad \mathbf{V} = \boldsymbol{a}_i^{(in)}\mathbf{W}^V \quad (13)$$

where $\mathbf{W}^Q$, $\mathbf{W}^K$ and $\mathbf{W}^V$ represent the mapping weights to the tuple of query, key and value. In multi-head attention (MHA), we concatenate the outputs of several self-attention modules and transform them by an affine operation. Without the loss of generality, we focus on single layer attention. Furthermore, we reorganize Equation (13) to fit the formulation of Equation (4):

$$\boldsymbol{z} = f_i^{(in)}([\mathbf{Q}, \mathbf{K}, \mathbf{V}]) := \text{softmax}\left(\frac{\mathbf{Q}\mathbf{K}^\top}{\sqrt{d_k}}\right)\mathbf{V}$$
$$[\mathbf{Q}, \mathbf{K}, \mathbf{V}] = \boldsymbol{a}_i^{(in)}\mathbf{W}_i^{(in)} := \boldsymbol{a}_i^{(in)}[\mathbf{W}^Q, \mathbf{W}^K, \mathbf{W}^V] \quad (14)$$

Equation (13) identifies the concrete definitions of $f^{(in)}$ and $\mathbf{W}_i^{(in)}$ for self-attention modules in the framework by Equation (4) so that we can plug these definitions employ Equation (5) to reconstruct the input of the attention layer by the leaked gradients.

Due to the large amount of parameters in ViT, we use Coarse-GIT for input reconstruction. If we use $\boldsymbol{g}^Q, \boldsymbol{g}^K, \boldsymbol{g}^V$ to represent the leaked gradients of $\mathbf{W}^Q$, $\mathbf{W}^K$ and $\mathbf{W}^V$, respectively, then we employ an MLP module $f$ parameterized by $\vartheta$ to reconstruct $\boldsymbol{a}_i^{(in)}$ in a self-attention module.

$$\boldsymbol{a}_i^{(in)} = f_\vartheta(\boldsymbol{g}^Q, \boldsymbol{g}^K, \boldsymbol{g}^V, \{\boldsymbol{g}_j^{(out)}\}_{j=1}^{N_{out}}, \boldsymbol{z}) \quad (15)$$

where $\{\boldsymbol{g}_j^{(out)}\}_{j=1}^{N_{out}}$ are the leaked gradients of output matrices as defined for a general MIMO layer in Section 3. The gradient inversion of fully-connected layers and residual structure in the ViT follows the same formulation as described in previous sections. Altogether, we can iteratively employ these formulas to reconstruct the input estimation of each layer, starting from the last layer and progressing to the first layer of the ViT model, eventually obtaining the input data estimation.

## G MORE EXPERIMENTAL ANALYSES AND ABLATION STUDIES

### G.1 DISCREPANCIES IN MODEL PARAMETERS

In federated learning, gradient sharing may be asynchronous Geiping et al. (2020), leading to slight discrepancies in model parameters across different nodes. Such inconsistencies can affect the performance of both optimization-based and generative reconstruction methods.

To create discrepancies in model parameters, we train each node with different volume of local dataset for several epochs, then we use generative models trained on input-gradient pairs from one node, i.e.,

the node under attack, to reconstruct the input data from another node, which is not necessarily under attack and has parameter discrepancies. In this context, larger local data and more training epochs lead to larger parameter discrepancies, making the input data reconstruction task more challenging.

Our experimental results under different settings are summarized in Table 7. We can clearly observe that GIT achieves the best performance under significant parameter discrepancies and is still capable of reconstructing high-quality training data, demonstrating its robustness to variations in model parameters across nodes. We also find that increasing the number of local datasets sometimes leads to reduced performance degradation. This may be because a larger number of local datasets increases the likelihood of including samples from classes that are easier to reconstruct, thus introducing a degree of randomness that favors recovery.

Table 7: Comparison of the MSE for GIT with varying parameter discrepancies. The parameter discrepancies is quantified by volume of local dataset & number of locally trained epochs. The leaked model for CIFAR10 is LeNet, and for ImageNet is ResNet.

| Dataset | Volume of Local Dataset | Method | Number of Locally Trained Epochs | | |
|---|---|---|---|---|---|
| | | | 0 | 10 | 20 |
| CIFAR10 | 500 | IG | 0.082 | 0.089 | 0.096 |
| | | LTI | 0.015 | 0.019 | 0.023 |
| | | GIT | **0.010** | **0.013** | **0.017** |
| | 1000 | IG | 0.082 | 0.097 | 0.102 |
| | | LTI | 0.015 | 0.026 | 0.030 |
| | | GIT | **0.010** | **0.017** | **0.020** |
| | 10000 | IG | 0.082 | 0.096 | 0.107 |
| | | LTI | 0.015 | 0.032 | 0.036 |
| | | GIT | **0.010** | **0.029** | **0.034** |
| ImageNet | 500 | IG | 0.161 | 0.162 | 0.162 |
| | | LTI | 0.043 | 0.049 | 0.053 |
| | | GIT | **0.039** | **0.043** | **0.044** |
| | 1000 | IG | 0.161 | 0.162 | 0.163 |
| | | LTI | 0.043 | 0.049 | 0.053 |
| | | GIT | **0.039** | **0.043** | **0.043** |
| | 10000 | IG | 0.161 | 0.164 | 0.164 |
| | | LTI | 0.043 | 0.048 | 0.051 |
| | | GIT | **0.039** | **0.040** | **0.040** |

## G.2 RECONSTRUCTION WITH CLIPPED GRADIENTS

The prune rate $\gamma$ represents the proportion of gradient directions with small absolute values that are pruned (Pruning is applied by a mask with 0 and 1 values, therefore the dimension of gradients is not changed). As shown in Table 8, gradient pruning has minimal impact on GIT's performance but significantly degrades the performance of DLG. The results indicate that even when pruning 90% of the gradient components with smaller absolute values, generative approaches remain largely unaffected, relying only on the top 10% of the largest gradient values for training. This suggests that generative approaches primarily capture the dominant gradient components with large absolute values during training, unlike optimization-based methods, which require a finer alignment with the full gradient information.

It can be deduced from Table 8 that generative approaches are less effective than optimization-based methods in recovering fine-grained image details. However, they demonstrate greater robustness against inaccurate gradients and gradient pruning while also being more efficient. Furthermore, generative methods train significantly faster, as gradient matching requires extensive computation to precisely align finer gradient details, leading to higher time complexity.

Table 8: Comparison of the MSE under gradient pruning with varying pune rate. The dataset is CIFAR10 and the leaked model is LeNet.

| Prune rate $\gamma$ | DLG | LTI | GIAS | GIT |
|---|---|---|---|---|
| 0 | 0.073 | 0.015 | 0.027 | **0.010** |
| 0.9 | 0.098 | 0.016 | 0.033 | **0.010** |
| 0.99 | 0.116 | 0.021 | 0.049 | **0.016** |
| 0.999 | 0.187 | 0.049 | 0.070 | **0.040** |

Table 9: MSE comparison with varying volumes of training data. The dataset is CIFAR10 and the leaked model is LeNet.

| Data Volume | LTI | GIAS | GIT |
|---|---|---|---|
| 1000 | 0.039 | 0.049 | **0.027** |
| 2000 | 0.033 | 0.045 | **0.021** |
| 5000 | 0.027 | 0.036 | **0.015** |
| 10000 | 0.015 | 0.027 | **0.010** |

### G.3 EFFECT OF NOISE ON THE PERFORMANCE OF OPTIMIZATION-BASED RECONSTRUCTION METHODS

Under inaccurate gradients, generative approaches demonstrate robust performance, as shown in Table 4. However, IG fails to recover meaningful data with as little as $0.01$ noise applied. This highlights the significant impact of noise on gradient matching methods like DLG. In the contrast, IG fails to recover meaningful data with as little as $0.01$ noise applied. This highlights the significant impact of noise on optimization-based methods like IG.

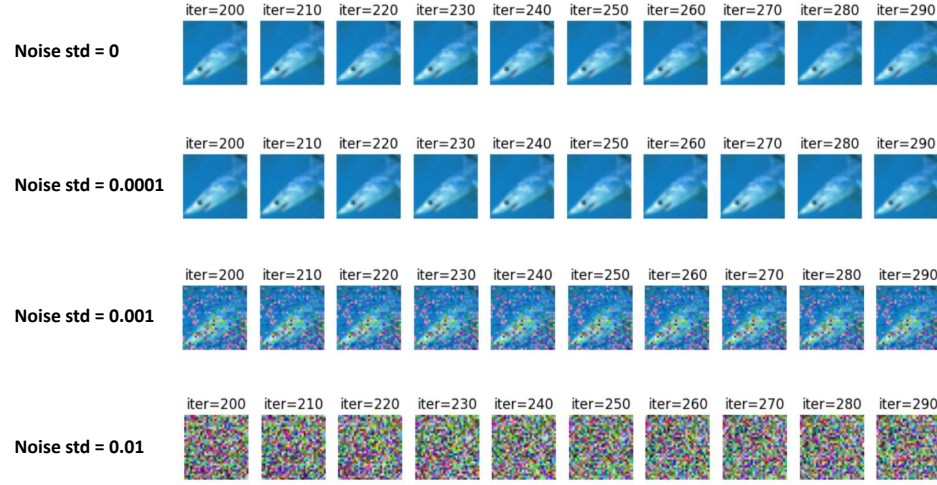

Figure 4: The figure illustrates the reconstructed images for IG when the leaked model is LeNet and the dataset is CIFAR-10. Varying levels of noise are applied to the gradients. The results depict IG's reconstructions between the 200th and 300th optimization iterations.

### G.4 RECONSTRUCTION WITH DIFFERENT VOLUMES OF TRAINING DATA

Training data volume refers to the size of the auxiliary dataset sampled by the attacker from public data. A larger training dataset, akin to performing data augmentation, can enhance the generalization ability of the generative model. However, it also incurs higher computational costs, requiring more time and resources to train the model. Moreover, in practice, acquiring a large volume of data with a distribution similar to the local dataset can be challenging. Given this tradeoff, it is essential to evaluate the performance of the generative approach under different training data volumes. In this section, we evaluate the performance of GIT using varying amounts of training data: $1,000$, $2,000$, $5,000$ and $10,000$ samples. Table 9 presents the impact of training data volume on the performance of the generative approach. It demonstrates that even with only $1,000$ input-gradient pairs, GIT is capable of reconstructing reasonable images, indicating that effective recovery is achievable with a limited amount of training data. As the generative model achieves near-perfect performance

with larger training sets, we can conclude that increased data volume helps mitigate overfitting and consequently improves model performance. Moreover, GIT consistently outperforms both LTI and GIAS across all settings.

## G.5 RECONSTRUCTION WITHOUT GRADIENT OF LAST LAYER'S BIAS

When the last layer of the neural network has a bias term $b_N$, i.e., $a_N = \mathbf{W}_N a_{N-1} + b_N$, following the idea of Ma et al. (2023), we have $\frac{\partial \mathcal{L}}{\partial z_N} = \frac{\partial \mathcal{L}}{\partial b_N}$. That is to say, we can directly utilize the gradient of the bias term in the last year as $\frac{\partial \mathcal{L}}{\partial z_N}$. When the last layer of the neural network does not have a bias term, we cannot directly obtain $\frac{\partial \mathcal{L}}{\partial z_N}$. Therefore, we employ ablation study using the leaked gradients for the weight of the last layers to estimate the output logits by a shallow MLP.

Table 10: Quantitative comparison for GIT with and w/o gradient of last layer's bias.

| Dataset | Leaked Model | Method | MSE↓ | PSNR↑ | LPIPS↓ | SSIM↑ |
|---------|--------------|--------|------|-------|--------|-------|
| CIFAR10 | LeNet | with bias | 0.010 | 20.38 | 0.2663 | 0.5533 |
|         |       | w/o bias | 0.012 | 19.35 | 0.2879 | 0.5035 |
| Imagenet | Resnet | with bias | 0.039 | 14.42 | 0.8513 | 0.3507 |
|          |        | w/o bias | 0.039 | 14.30 | 0.9017 | 0.3120 |

## G.6 OPTIMIZATION-BASED METHODS WITH AND W/O GENERATED IMAGE PRIOR

As shown in the Figure 5, the blue curve represents the convergence of the hybrid method, while the red curve illustrates IG without an image prior. It is evident that the hybrid method not only converges faster but also achieves superior performance. The fluctuations in the blue convergence curve are due to the small learning rate set for the optimizer, which causes oscillations when the loss falls below 1.

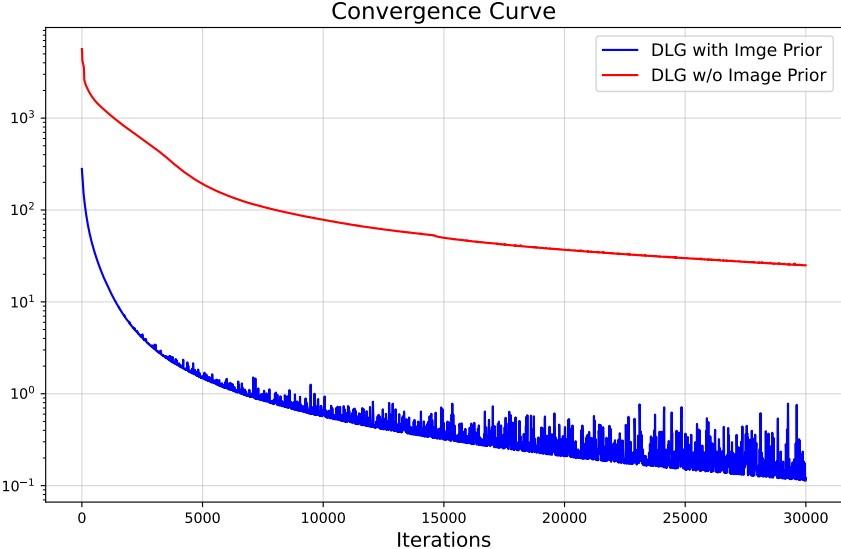

Figure 5: The convergence curve of DLG with and without an image prior. The leaked model is ResNet. The vertical axis indicates the distance between the dummy gradients and the corresponding ground-truth gradients.

## G.7 VISUAL RESULTS

### G.7.1 VISUAL RESULTS ON CIFAR-10 AND TINY IMAGENET

Figure 6 illustrates reconstructed CIFAR-10 and Tiny ImageNet direct using GIT for reconstruction or using GIT as generated image prior. These results show that directly using GIT for reconstruction can achieve reasonable recovery but tends to lose some high-frequency details. In contrast, using GIT as an image prior—specifically for initializing optimization-based methods—helps preserve high-frequency information and achieves reconstruction quality beyond what optimization-based methods alone can attain.

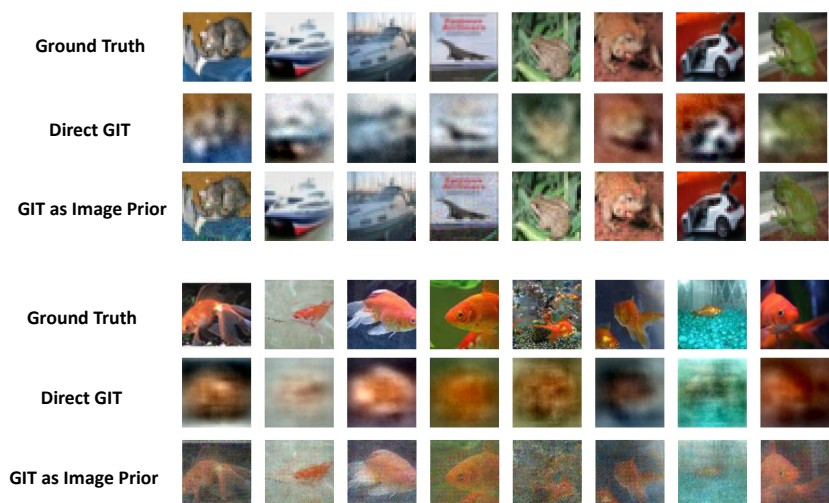

Figure 6: The figure illustrates the **ground truth input images**, the **direct reconstructed images by GIT** and the **reconstructed images using IG initialized with GIT-generated prior**, from top to the bottom respectively. The top three rows correspond to the CIFAR-10 dataset with the leaked model being LeNet, while the bottom three rows correspond to the TinyImageNet dataset with the leaked model being ResNet.

### G.7.2 VISUAL RESULTS FOR GIT ON LARGE RESOLUTION

Reconstruction at high resolution tends to be more challenging, especially for images containing complex objects. To illustrate the characteristics of both easy and hard-to-recover examples, we present the first 8 and the best 100 reconstructions. The results are shown in Figure 7 and Figure 8. Odd-numbered rows show the ground-truth images, while even-numbered rows display the corresponding reconstructions obtained directly using GIT.

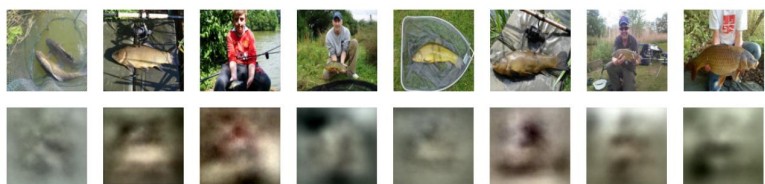

Figure 7: The first 8 reconstructed images (ImageNet, ResNet). Odd-numbered rows show the ground-truth images, while even-numbered rows display the corresponding reconstructions obtained directly using GIT.

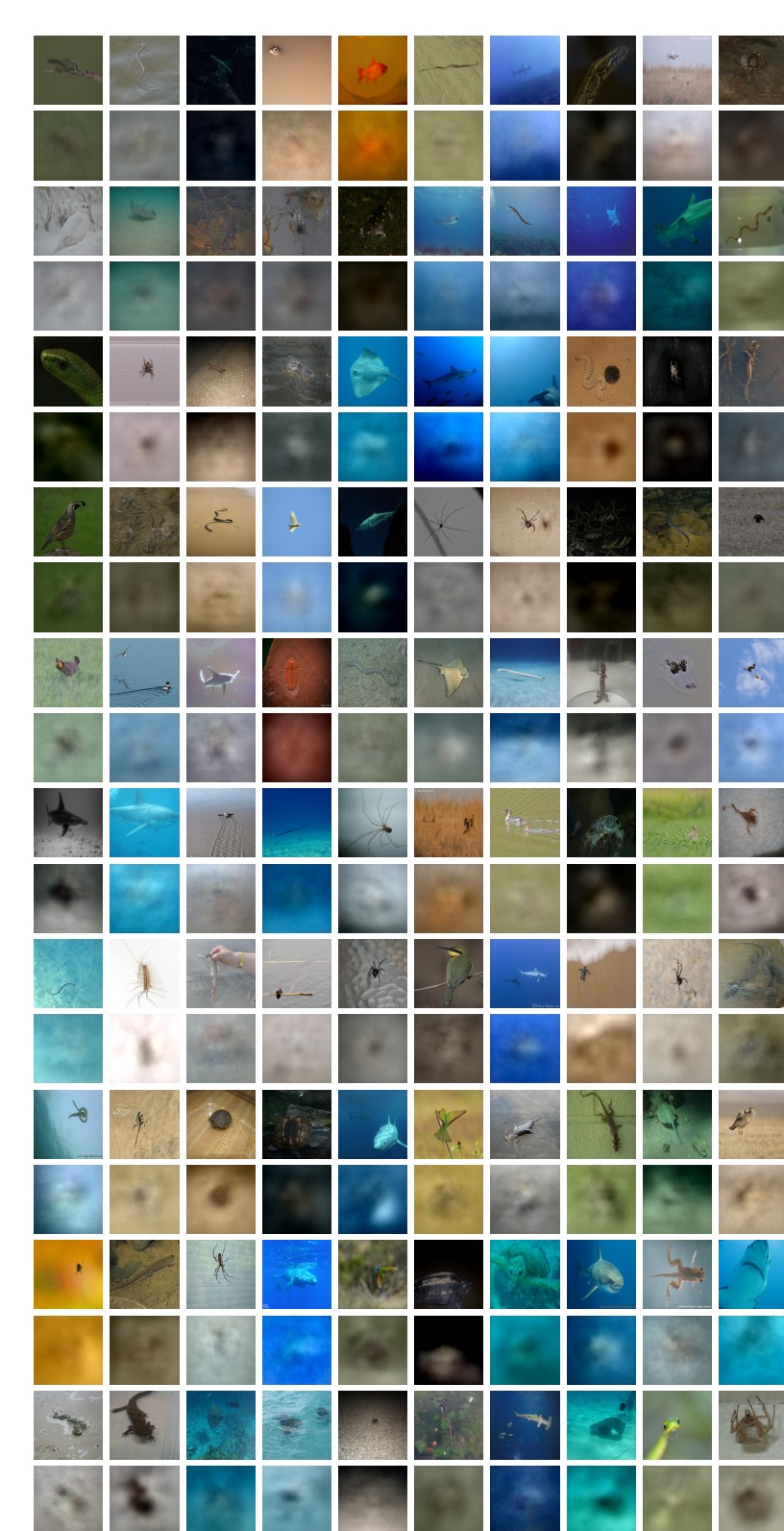

Figure 8: The best 100 reconstructed images with lowest MSE (ImageNet, ResNet). Odd-numbered rows show the ground-truth images, while even-numbered rows display the corresponding reconstructions obtained directly using GIT.

