# OpenReview forum: "Gradient Inversion Transcript: Leveraging Robust Generated Priors to Reconstruct Training Data from Gradient Leakage"
_ICLR.cc/2026/Conference — Submitted to ICLR 2026_

### Official Review · Reviewer_xnwp · 2025-10-24

**Soundness:** 1
**Presentation:** 1
**Contribution:** 2
**Rating:** 2
**Confidence:** 4

**Summary:**

This paper introduces Gradient Inversion Transcript (GIT), a novel model-based method for reconstructing training data from leaked gradients in federated learning. The core idea is to construct a reconstruction model whose architecture is adaptively tailored to approximate the inverse of the target model's back-propagation process. The authors propose two variants: a theoretically-grounded "Exact-GIT" that directly implements the derived recursive inversion formula, and a more practical "Coarse-GIT" that uses MLPs to approximate this process. The paper claims that GIT is more efficient and robust than existing methods, showing strong performance across various datasets and under challenging conditions like gradient perturbations and distribution shifts. Furthermore, the output of GIT is proposed as a prior to accelerate and improve optimization-based attacks.

**Strengths:**

1. The idea of constructing an inversion process by modelling the recursive relationship of layer input gradients through layer-by-layer insertion of the corresponding gradients is creative and is a potentially valuable improvement over existing work. The mathematics is also technically sound.

2. Evaluating GIT not only as a standalone attack, but in combination with existing optimization methods is a good contribution towards refining different frameworks.

3. Under the evaluated settings, it seems that GIT is substantially more effective compared to existing attacks.

**Weaknesses:**

1. **Mathematical writing and details**: The details in Section 3 are unclear and contain factually wrong mathematics:
   - For the most common applications in this field, the gradient operator on a matrix is defined as the element-wise operator on each gradient element. In their definition, the authors instead seem to use the transpose, causing slight confusion in the manuscript. They should clarify this to avoid inconsistencies between different readers' background knowledge.
   - Tensor multiplication is usually a different operation to what is defined in the paper (i.e. in the standard tensor multiplication definition, a $A\times B$ matrix multiplied by a $C\times D$ matrix will result in a $A\times B \times C\times D$ tensor). I believe what the authors denote is a simple matrix multiplication, and should clarify this to avoid further confusion.
   - The authors should either present proofs of their statements in Appendix, or refer to existing proofs in the main text, as the results are non-trivial. In the current state the writing level is below par to what is expected in an ICLR submission.

2. **Writing and Clarity**:
   - I elaborate on clarity issues in the Introduction and Related Work sections in **W3** and **W4**.
   - The technical details are overall too vague across the entire paper. A rewrite with separate sections/paragraphs in the main text or Appendix are warranted for clarifying the attacker assumptions, the mathematical derivations, algorithmic description, and the experimental settings. It must be clarified where the derivations in Section 3 are used as inspiration, and where they are directly incorporated as part of the inversion MLP.
   - It is unclear how much of the Section 3 derivations are equivalent or inspired by R-GAP [5], who perform a similar recursive approach explicitly on a layer-by-layer basis.
   - (Minor) I would discourage the authors from referencing prior work only through hyperlinks in Table 1, without explicit references in the main text (as they did for iLRG and SPEAR), and acknowledging the difference from other related work.
   - (Nitpick) In L430 the authors refer to themselves as "I".

3. **Setting specification**: The authors seem to present a generally unorthodox problem setting. First, in their setting, the authors describe:
    - "Our method does not need the parameters of the leaked model or the labels of the training data, as GIT trains the auxiliary reconstruction model on publicly available data with known labels." -> Given that in order to obtain gradients for the training data, one needs to assume knowledge of the model weights, this statement's assumptions are inherently stronger than acknowledged.
    - " As illustrated in Figure 1, we adopt a similar premise to DLG Zhu et al. (2019): the attacker hacks the channel to inject data to one client, which shares the gradient with the server and other clients" -> This statement is incorrect and confusing for multiple reasons: 1) To the extent of my knowledge, no data is injected in any gradient leakage setting, and the authors describe no such event in their paper as well, 2) the correlation between data injection and gradient sharing is unclear and seemingly unrelated, 3) "Hacking" a single client does not seem to relate to the other clients, who most often share their gradients with the server, 4) In the most usual case (including DLG) of gradient inversion, no hacking is involved, but a server, that can be either malicious, or honest-but-curious, monitors the shared gradients and provides the model updates to the clients. In some distributed settings, some of these interactions are slightly different, but the core assumptions remain the same.
    - Figure 1 introduces the notion of a hacked channel, which can both both somehow inject data and receive gradient information. This seems more unrealistic than the honest-but-curious server setting, as it assumes that the client is is completely unaware of the data they are using, as well as the gradients they are sending. This proposal seems like an unnecessary man-in-the-middle-like attack, which also does not interact with other clients, despite the claims of the authors that the attack will reconstruct both the hacked user (who should be using the injected data?), and the remaining users' data.
    - "The attack aims to reconstruct the data from both the hacked client and other clients by shared gradients" -> This statement introduces the notion of a "hacked client", without emphasis on what the hacking entails or what actions can be read/written.
    - "unlike iterative optimization-based methods, which are not applicable for scenario (2) since the attacker has no access to inject data to unhacked clients and is therefore unable to reconstruct training data from them." -> The injection angle is again unclear, and the authors must clarify their assumptions regarding this again.

 The paper fails to articulate a coherent or realistic threat model. The 'hacked channel' and 'data injection' premises are contrived and deviate from established literature without justification, while the claim of reconstructing data from non-hacked clients remains entirely unsubstantiated.

4. **Related work**: I remain unconvinced of the familiarity of the authors with related work, leading to the confusing and questionable position of GIT in the gradient leakage field. I write some comments and corrections below:
    - **RTNN**, while also dealing with privacy, is a **model-inversion** technique, which explores a significantly different challenge to gradient leakage in Federated Learning.
    - Most of the settings, described in Table 1 are wrong. For one, essentially all gradient leakage attacks **assume knowledge of the model parameters and architectures**. This is because plenty of them rely on optimising the proxy gradients to be as close as possible to the observed ones, for which you would need to emulate both the backward and forward pass. This is the fundamental assumption of **DLG [1], Geiping et al. [2], iDLG [3], etc.**. This is what I suppose the authors refer to as "the gradient query" pipeline, but having access to that and having the access to the model are essentially equivalent.
    - As some model-based methods also incorporate such a loss, that also holds for that section as well.
    - The column "Pretrained GAN" restricts the architecture of the generative model without no particular purpose, i.e. DGGI [4] uses a diffusion model.
    - SPEAR [5] 's position in the field is different than what the authors have described. The attack requires the existence of a ReLU-activated linear layer, but not much beyond that. In particular, the authors specifically highlight in the paper that their method does not require access to the data labels, or whatever the authors refer to as the "output logit", which should be further clarified.
    - LIT, and by extension, GIT, also require the computation of gradients for their training set. This, by extension, assumes knowledge of the model weights and architecture.

 This misrepresentation of prior work invalidates the paper's comparative analysis and suggests the authors do not have a firm grasp of the literature. The current state of the introduction and related work sections warrants a full rewrite of the transcript, without which this paper will remain scientifically invalid.

5. **Exact-GIT**: The authors present 2 methods, with **Exact-GIT** supposed to more closely emulate Equations (3) and (5)'s explicit formulae. However, the only results the authors present is an analysis into the closeness of the trained parameters with the actual model weights. Given that a substantial part of the paper is dedicated to the recurrent relationship methodology, it is unclear whether this contribution is valid without any reconstruction results. Further, given the lack of baseline or explanation of the values, it is further unclear what the trainable parameters starting to approach the model weights implies. Lastly, the authors describe in L244 that they treat the "unknown weights" as additional parameters (which is in relevant settings unnecessary, as described in **W3**), making the system underdetermined, with their being the same number of weight parameters, as gradient constraints, on top of which there are also input free variables.

6. **Evaluation**:
    - All evaluation settings omit specifying the batch size and image size, which limits the understanding of the difficulty of the setting.
    - The provided reconstruction images in G.7 seem quite blurry, while the prior work the authors compare to usually include images of significantly better quality. This discrepancy puts the results and practicality of the attack into question.
    - The "ViT" setting is not elaborated upon, particularly how the architecture relates to the transformer one and Equation 5. The authors must be more precise in their explanations.
    - The "Innacurate Gradients" setting is usually supported by reporting the accuracy on the main task, so as to show that the reported noise is not too disruptive to the training pipeline, which the auhtors hav enot provided.
    - The "Distribution shift" experiments 1) use overlapping classes in basically all settings, and 2) still contain data from the same dataset, which may have a particular type of prior distribution behind them. This weakens the claim that GIT works under significant distribution shifts.

**Questions:**

1. What is the setting, in which GIT operates? The authors should clarify all assumptions for the attacker and system explicitly and clearly.

2. The authors must provide further experimental details.
    - What batch size was used for each experiment? Assuming a batch size of 1, the authors only evaluate against outdated attacks, while more modern attacks have been shown to handle batch sizes significantly larger than 1.
    - How does the attack perform without having a method to analytically compute the output logits, especially for higher batch sizes?
    - If, as claimed in the intro, the attack "reconstructs the data from both the hacked client and other clients", what is the difference between the settings, and what is the performance difference?
    - The authors claim in 5.2.1 that "The results confirm the vulnerability of optimization-based methods against gradient perturbations". However, outside of IG, the other attacks remain seemingly unaffected by the perturbations, suggesting either that the authors' claim is incorrect, or there exists a flaw in the evaluation. Can the authors explain this discrepancy?

3. What are the reconstruction results for Exact-GIT, and how is the weight convergence relevant to them?

4. The supplementary material provides an implementation for the MLP and ResNet variation of what seems like Coarse-GIT, there seem to be no details on the setup for Exact-GIT. Given that most of theoretical development is unused in Coarse-GIT, not having the option to investigate and replicate the Exact-GIT implementation is unfortunate. Can the authors provide additional clarification, and add the implementation to the supplementary material?

5. The authors report lower PSNR for Geiping et al. [2] for CIFAR10 with the LeNet network than the original paper, even compared to batch sizes of 1, posing the evaluation methodology into question. Why is that the case?

6. As training goes on, the model weights change, and hence the gradients of any input will change in comparison to the initial model state. How does a GIT, trained on the initial model state, perform over the FL training period?

7. Can the authors compare their proposed methodology to that of R-GAP? Given that both techniques propose methods based on similar recursive formulae, the authors should make this clear.

8. Given that Equation 3.2 does not depend on $f_i^{out}$, can the authors discuss the implications of this finding? This seems to imply that depending on how one picks $f^{in}$ and $f^{out}$, it might result in a system of varying difficulty.

9. The authors should address all remaining concerns from the **Weaknesses** section.

## Current recommendation

I am assinging this paper a recommendation of **2: Reject**. Despite the creative central concept of formally inverting the back-propagation process, this paper suffers from fundamental flaws that render it unsuitable for publication in its current state. The problem setting is unorthodox and poorly justified, with confusing and unrealistic assumptions about a "hacked client" and "data injection" that deviate from standard gradient leakage scenarios, as well as the paper's own application. Critically, the mathematical derivations contain significant errors, most notably in Equation 5, and lack the necessary rigor, invalidating the conclusions that follow. The related work section and comparison table demonstrate a profound misunderstanding of prior literature, misrepresenting the core assumptions. Combined with a lack of empirical results for the "Exact-GIT" method and insufficient detail in the experimental setup, the paper's claims are not adequately supported. A complete overhaul of the problem formulation, mathematical foundations, and literature review is required. I encourage the authors to address the weaknesses and questions during the discussion phase, so they can clarify some of my concerns and improve their transcript.

### References

[1] Zhu, Ligeng, Zhijian Liu, and Song Han. "Deep leakage from gradients." Advances in neural information processing systems 32 (2019).

[2] Geiping, Jonas, et al. "Inverting gradients-how easy is it to break privacy in federated learning?." Advances in neural information processing systems 33 (2020): 16937-16947.

[3] Zhao, Bo, Konda Reddy Mopuri, and Hakan Bilen. "idlg: Improved deep leakage from gradients." arXiv preprint arXiv:2001.02610 (2020).

[4] Dimitrov, Dimitar I., et al. "Spear: Exact gradient inversion of batches in federated learning." Advances in Neural Information Processing Systems 37 (2024): 106768-106799.

[5] Zhu, Junyi, and Matthew Blaschko. "R-gap: Recursive gradient attack on privacy." arXiv preprint arXiv:2010.07733 (2020).

---

> ### Author Response · Authors · 2025-11-21
>
> ## Weakness 1:
> Mathematical writing and details: The details in Section 3 are unclear and contain factually wrong mathematics.
> ## Response to Weakness 1:
> Based on the reviewer’s comments regarding **tensor-level multiplication**, **applying transposes $(\cdot)^T$ directly to tensors**, and the **confusion in the notation of $\otimes$**, we have revised and clarified our definitions. Here we define three operators for 3rd-order tensors (batch × features × features):
>
> > $\otimes$: Tensor multiplication. For two 3-d tensors $\mathbf{A} \in \mathbb{R}^{B \times d_1 \times d_2}$ and $\mathbf{B} \in \mathbb{R}^{B \times d_2 \times d_3}$, , $\mathbf{A} \otimes \mathbf{B} \in \mathbb{R}^{B \times d_1 \times d_3}$ is defined as the batch-wise matrix multiplication along the last two dimensions for each batch element **for each batch element independently**, ignoring the batch dimension. Formally, for each $i \in \{1, \dots, B\}$:
> $(\mathbf{A} \otimes \mathbf{B})[i, :, :] = \mathbf{A}[i, :, :] \times \mathbf{B}[i, :, :]$, where $\times$ denotes standard matrix multiplication.
> Note that for any second-order matrix such as  $W_j^{(\text{out})} \in \mathbb{R}^{d_j^{(\text{out})} \times d}$, we broadcast it into a **third-order tensor** of shape $1 \times d_j^{(\text{out})} \times d$, so that the first dimension (the batch dimension) is aligned with the other tensors during the tensor operations.
>
> > $\odot$: Broadcast row-wise product. For $\mathbf{A} \in \mathbb{R}^{B \times d_1 \times 1}$ and $\mathbf{B} \in \mathbb{R}^{B \times 1 \times d_2}$, $\mathbf{A} \odot \mathbf{B} \in \mathbb{R}^{B \times d_1 \times d_2}$ is defined element-wise as $(\mathbf{A} \odot \mathbf{B})[i,j,k] = \mathbf{A}[i,j,1] \cdot \mathbf{B}[i,1,k]$.
>
> > $(\cdot)^T$: Transpose operator. For a tensor $\mathbf{X} \in \mathbb{R}^{B \times d_1 \times d_2}$, $\mathbf{X}^T \in \mathbb{R}^{B \times d_2 \times d_1}$ swaps the second and third dimensions for each batch element.
>
> Moreover, in response to the reviewer’s concern about unclear derivations, we provide **detailed derivations of Equations (2) and (3)**:
>
> $$
> \mathbf{z} = \sum_{i=1}^{N_{\text{in}}} W_i^{(\text{in})} \cdot \mathbf{a}_i^{(\text{in})}
> $$
>
> $$
> \mathbf{a} = \sigma(\mathbf{z})
> $$
>
> $$
> \mathbf{z}_j^{(\text{out})} = W_j^{(\text{out})} \cdot \mathbf{a}
> $$
>
> $$
> \mathbf{a}_j^{(\text{out})} = \sigma_j(\mathbf{z}_j^{(\text{out})})
> $$
>
> Forward propagation complete. Next, compute gradients with respect to weights:
>
> $$
> \delta_j^{(\text{out})} = \frac{\partial \mathcal{L}}{\partial \mathbf{z}_j^{(\text{out})}}
> = \frac{\partial \mathcal{L}}{\partial \mathbf{a}_j^{(\text{out})}} \odot \sigma_j'(\mathbf{z}_j^{(\text{out})})
> $$
>
> $$
> \mathbf{g}_j^{(\text{out})} = \frac{\partial \mathcal{L}}{\partial W_j^{(\text{out})}}
> = \delta_j^{(\text{out})} \otimes \mathbf{a}^T
> $$
>
> Backpropagation to MIMO layer:
>
> $$
> \delta = \sum_{j=1}^{N_{\text{out}}} (W_j^{(\text{out})})^T \otimes \delta_j^{(\text{out})} \odot \sigma'(\mathbf{z})
> $$
>
> Gradient w.r.t. input weights:
>
> $$
> \mathbf{g}_i^{(\text{in})} = \frac{\partial \mathcal{L}}{\partial W_i^{(\text{in})}}
> = \delta \otimes (\mathbf{a}_i^{(\text{in})})^T
> $$
>
> Approximate reconstruction of input:
>
> $$
> \delta \approx \mathbf{g}_i^{(\text{in})} \otimes (\mathbf{a}_i^{(\text{in})T})^+
> $$
>
> $$
> \delta \odot (\sigma'(\mathbf{z}))^+ \approx \sum_{j=1}^{N_{\text{out}}} W_j^{(\text{out})T} \otimes \mathbf{g}_j^{(\text{out})} \otimes (\mathbf{a}^T)^+
> $$
>
> Combine to solve for $(\mathbf{a}_i^{(\text{in})})^T$:
>
> $$
> (\mathbf{a}_i^{(\text{in})})^T \simeq \left( \Sigma_j W_j^{(\text{out})T} \otimes \mathbf{g}_j^{(\text{out})} \otimes (\mathbf{a}^T)^+ \odot (\sigma'(\mathbf{z})) \right)^+ \otimes \mathbf{g}_i^{(\text{in})}
> $$

---

> > ### Author Response · Authors · 2025-11-21
> >
> > ## Weakness 2.2:
> > The technical details are overall too vague across the entire paper.
> > ## Response to weakness 2.2:
> > In short, our approach reconstructs data **layer-wise** or **module-wise**, with each layer/module corresponding to one MLP whose input is determined by the theoretical formulas. The GIT generator is built iteratively, and due to its **high-level architecture adaptability**, no single general formula can fully describe it. The iterative construction is detailed in **Algorithm 1**.
> > ## Weakness 2.3:
> > It is unclear how much of the Section 3 derivations are equivalent or inspired by R-GAP.
> > ## Response to weakness 2.3:
> > Our method, similar to R-GAP, is **backpropagation-inspired**, but the two approaches differ fundamentally in their **assumptions**, which lead to different methodologies.
> > R-GAP operates in a **white-box setting** where the model parameters are fully known.
> > In contrast, GIT assumes a **black-box setting** in which only gradients are accessible while the model parameters remain unknown. Under this assumption, **exact-GIT** uses the gradient information to estimate the unknown weight matrices that R-GAP would directly use.
> >
> > **Coarse-GIT** differs from exact-GIT in that it does **not** explicitly estimate the weight matrices. Instead, it trains a small MLP for each module of the target model—for example, a layer in LeNet or ResNet, or a multi-head attention block in a ViT. These MLPs are connected according to the original network architecture, preserving structures such as residual blocks.
> >
> > Overall, our method can be viewed as a **black-box counterpart of R-GAP**, achieving similar reconstruction behavior under substantially weaker assumptions.
> > ## Weakness 3.1:
> > Setting specification: Given that in order to obtain gradients for the training data, one needs to assume knowledge of the model weights, this statement's assumptions are inherently stronger than acknowledged.
> > ## Response to weakness 3.1:
> > The reviewer may not have considered the commonly used **black-box setting**. In black-box scenarios such as membership inference attacks, the attacker typically only has access to model outputs such as gradients or confidence scores, while the model weights remain unknown.
> > ## Weakness 3.2:
> > Setting specification: confusion about data injection, client hacking and gradient sharing.
> > ## Response to weakness 3.2:
> > Gradient leakage refers to the **gradient-sharing** mechanism in FL systems—this happens inherently during FL training and is **not an attack** by itself. In contrast, **data injection, honest-but-curious servers, malicious datasets, and malicious servers** are all parallel forms of attack settings (as summarized in **SoK: Gradient Inversion Attacks in Federated Learning**). These different attack settings can all be applied to FL systems; for example, backdoor attacks operate by injecting data and do not require access to the model.
> >
> > Our black-box setting assumes less information than the honest-but-curious server setting, which is a white-box problem. Since our scenario is strictly more constrained, **any method that works under our black-box assumption also naturally satisfies the requirements of the white-box setting**.
> > ## Weakness 3.3:
> > Setting specification: injecting data and receiving gradient information seems more unrealistic than the honest-but-curious server setting.
> > ## Response to weakness 3.3:
> > As stated in our response to Weakness 3.2, all of these attack settings are validated and discussed in SoK: Gradient Inversion Attacks in Federated Learning.
> >
> > Regarding the reviewer’s concern about our figure, we acknowledge that there are indeed issues. To ensure that the hacked client can obtain the shared gradients, the system should either follow a **decentralized FL framework** or use a **broadcast mechanism in a centralized FL** setup. We will correct this in the revised PDF.
> > ## Weakness 3.4:
> > Setting specification: what the hacking entails or what actions can be read/written.
> > ## Response to weakness 3.4:
> > To summarize the intended workflow: clients share gradients with each other; one client is compromised by the attacker; the attacker can inject data into this black-box client—analogous to standard backdoor or data-poisoning attacks—while only having access to the gradients shared during FL training.
> > ## Weakness 3.5:
> > Setting specification: the injection angle is again unclear, and the authors must clarify their assumptions regarding this again.
> > ## Response to weakness 3.5:
> > Please refer to the answers to weakness 3.1 to 3.4.

---

> > > ### Author Response · Authors · 2025-11-21
> > >
> > > ## Weakness 4:
> > > Related work: I remain unconvinced of the familiarity of the authors with related work, leading to the confusing and questionable position of GIT in the gradient leakage field.
> > > ## Response to weakness 4:
> > > Due to space limitations, the Related Work section was kept relatively concise. In the revised PDF, we will provide a more detailed discussion of prior methods and clearly articulate how our approach differs from them.
> > > Our main motivation follows the explanation provided in our **response to Weakness 2.2**. For completeness, we will restate and elaborate on this motivation in the revised PDF.
> > > ## Weakness 5:
> > > Exact-GIT: it is unclear what the trainable parameters starting to approach the model weights implies. Lastly, the authors describe in L244 that they treat the "unknown weights" as additional parameters (which is in relevant settings unnecessary, as described in W3), making the system underdetermined.
> > > ## Response to weakness 5:
> > > Indeed, only Figure 3 in Appendix F.2 reports results for Exact-GIT, and this result illustrates how Exact-GIT operates in the **stricter black-box setting**. Specifically, **Exact-GIT explicitly** estimates the unknown ground-truth weights, whereas **Coarse-GIT relies on implicit** weight modeling through trainable MLP modules. In both Exact-GIT and Coarse-GIT, GIT ultimately performs an **implicit mapping from gradients to data** through the trained GIT generator, but the role of parameter reconstruction is fundamentally different between the two settings.
> > >
> > > The motivation for Exact-GIT is solely to demonstrate the **feasibility** of performing explicit weight estimation in a black-box scenario, even though this comes with well-known numerical instability due to the use of pseudo-inverse operations. Introducing trainable parameters $W$ in Exact-GIT does not make the system underdetermined. Rather, as we explained earlier, this design tightens the problem setting and shows how an attack can be constructed in a more stringent black-box scenario where the model parameters are entirely unknown.
> > > ## Weakness 6.1:
> > > All evaluation settings omit specifying the batch size and image size.
> > > ## Response to weakness 6.1:
> > > The batch size is fixed at 1 in the experiments, as indicated in the **caption of Table 4**. The image sizes correspond to the respective resolutions of the datasets used for different tasks—namely, CIFAR-10, ImageNet-100, FER, and JAFFE.
> > > ## Weakness 6.2:
> > > The provided reconstruction images in G.7 seem quite blurry, while the prior work the authors compare to usually include images of significantly better quality.
> > > ## Response to weakness 6.2:
> > > Our method does underperform current generative model-based approaches in terms of reconstruction quality — but this is an unfair comparison. Generative model–based methods, including diffusion models and GANs, require pretraining on the target reconstruction dataset to obtain additional prior information, which makes a direct comparison with our proposed method unfair. For this reason, we compare against a generative model–based approach (GIAS **without pretraining**) in **Table 3**, which demonstrates that under fair comparison conditions — without **relying on pretrained generative models** — these methods are actually less effective than ours.
> > >
> > > **Our strengths** lie in **efficiency and robustness to adversarial perturbations**. All optimization-based and generative model-based methods require iterative optimization, which makes them highly vulnerable to gradient perturbations and extremely time-consuming even for a single batch. As a result, these methods are **impractical for reconstructing large volumes of images**. In contrast, gradient-input mapping remains highly **effective at scale** and demonstrates strong **resistance to noise** due to its direct mapping nature, as demonstrated in **Section 5.2.1**.
> > > ## Weakness 6.3:
> > > The "ViT" setting is not elaborated upon.
> > > ## Response to weakness 6.3:
> > > The architectural details of ViT and all experimental configurations are provided in **Appendix D**.
> > > ## Weakness 6.4:
> > > The "Innacurate Gradients" setting is usually supported by reporting the accuracy on the main task, so as to show that the reported noise is not too disruptive to the training pipeline.
> > > ## Response to weakness 6.4:
> > > We will add the **utility** section of the main tasks in the revised PDF.
> > > ## Weakness 6.5:
> > > The "Distribution shift" experiments 1) use overlapping classes in basically all settings, and 2) still contain data from the same dataset, which may have a particular type of prior distribution behind them.
> > > ## Response to weakness 6.5:
> > > We have also evaluated our method on completely different datasets — specifically, FER and JAFFE — as shown in **Table 6**.

---

> > > > ### Author Response · Authors · 2025-11-21
> > > >
> > > > ## Question 1:
> > > > What is the setting, in which GIT operates? The authors should clarify all assumptions for the attacker and system explicitly and clearly.
> > > > ## Answer to Question 1:
> > > > Please refer to the **responds to weakness 3.1 to 3.4**.
> > > > ## Question 2:
> > > > The authors must provide further experimental details.
> > > > ## Answer to Question 2:
> > > > All experimental configurations are provided in **Appendix D**, including FL model architectures and GIT generator architectures.
> > > > ## Question 3:
> > > > What batch size was used for each experiment? Assuming a batch size of 1, the authors only evaluate against outdated attacks, while more modern attacks have been shown to handle batch sizes significantly larger than 1.
> > > > ## Answer to Question 3:
> > > > The batch size is fixed at 1 in the experiments, as indicated in the **caption of Table 4**.
> > > > **Our strengths** lie in **efficiency and robustness to adversarial perturbations**, not in high resolution or large bath size. All optimization-based and generative model-based methods require iterative optimization, which makes them highly vulnerable to gradient perturbations and extremely time-consuming even for a single batch. As a result, these methods are **impractical for reconstructing large volumes of images**. In contrast, gradient-input mapping remains highly **effective at scale** and demonstrates strong **resistance to noise** due to its direct mapping nature, as demonstrated in **Section 5.2.1**.
> > > > ## Question 4:
> > > > How does the attack perform without having a method to analytically compute the output logits, especially for higher batch sizes?
> > > > ## Answer to Question 4:
> > > > Please refer to the answers to weakness 3.1.
> > > > ## Question 5:
> > > > If, as claimed in the intro, the attack "reconstructs the data from both the hacked client and other clients", what is the difference between the settings, and what is the performance difference?
> > > > ## Answer to Question 5:
> > > > We assume a **black-box setting**, where the attacker can only inject data into the hacked client but **has no access to the client's parameters or private data**. This is similar to backdoor-attack scenarios in which the adversary only interacts with the model through its data-input interface, without accessing the model itself. This setting is significantly more challenging than a white-box attack.
> > > >
> > > > The difference between the hacked client and other clients is that the GIT generator is trained using the model obtained from the hacked client together with a public dataset. When reconstructing data from other clients, the attack suffers from **parameter discrepancy**, because in federated learning different clients maintain slightly different versions of the shared model. In addition, the other clients’ gradients come from decentralized FL gradient sharing or centralized FL gradient broadcasting, which further contributes to the mismatch.
> > > > ## Question 6:
> > > > The authors claim in 5.2.1 that "The results confirm the vulnerability of optimization-based methods against gradient perturbations". However, outside of IG, the other attacks remain seemingly unaffected by the perturbations, suggesting either that the authors' claim is incorrect, or there exists a flaw in the evaluation. Can the authors explain this discrepancy?
> > > > ## Answer to Question 6:
> > > > The statement “confirm the vulnerability of optimization-based methods” **specifically refers to optimization-based attacks such as DLG and IG**. These methods do **not use an auxiliary model** (e.g., generative-model-based methods rely on a generative model, and input-gradient-mapping-based methods rely on a mapping generator). Instead, they depend purely on iterative gradient matching, which makes them highly sensitive to even small perturbations in the ground-truth gradients and therefore not robust.
> > > >
> > > > In contrast, attacks with a trained generator do not rely on direct gradient optimization, so their performance remains largely unaffected. Thus, the results are consistent with the claim.

---

> > > > > ### Author Response · Authors · 2025-11-21
> > > > >
> > > > > ## Question 7:
> > > > > What are the reconstruction results for Exact-GIT, and how is the weight convergence relevant to them?
> > > > > ## Answer to Question 7:
> > > > > Please refer to the **answer to weakness 5**.
> > > > > ## Question 8:
> > > > > Given that most of theoretical development is unused in Coarse-GIT, not having the option to investigate and replicate the Exact-GIT implementation is unfortunate.
> > > > > ## Answer to Question 8:
> > > > > We will provide a more detailed description of the model design in the appendix. It is also important to clarify that, as shown in **Algorithm 1**, Coarse-GIT does **not** discard the core theoretical development. Instead, it simply replaces the right-hand side of **Equation (3)** or **Equation (5)** with an MLP. (Equations (3) and (5) model each module—e.g., each layer of a ResNet or LeNet, or each multi-head attention block of a ViT—as an MLP, as stated in Lines 280–282.)
> > > > >
> > > > > Moreover, we train an MLP for each **module** of the target model: a layer in LeNet, a layer in ResNet, or a multi-head attention block in ViT. Our proposed method also adapts to **high-level architecture** changes: The connections between these MLPs follow the original model structure, including components such as residual blocks.
> > > > > ## Question 9:
> > > > > The authors report lower PSNR for Geiping et al. [2] for CIFAR10 with the LeNet network than the original paper, even compared to batch sizes of 1, posing the evaluation methodology into question. Why is that the case?
> > > > > ## Answer to Question 9:
> > > > > This is because we found that the reconstruction performance of DLG and IG is highly sensitive to architectural changes, particularly to the relationship between input and output dimensions. In our configuration (see **Appendix D**), we tested different kernel sizes and step sizes, and under the settings we used, the performance of DLG and IG degraded significantly. In contrast, our method remained unaffected.
> > > > > ## Question 10:
> > > > > As training goes on, the model weights change, and hence the gradients of any input will change in comparison to the initial model state. How does a GIT, trained on the initial model state, perform over the FL training period?
> > > > > ## Answer to Question 10:
> > > > > For different stages of the federated learning process, we need to train **a corresponding GIT generator**. This is because GIT performs the inversion of backpropagation, which is inherently affected by the leaked model. Existing work (CENSOR: Defense Against Gradient Inversion via Orthogonal Subspace Bayesian Sampling) has shown that as training progresses, all gradient inversion attacks become increasingly difficult. Therefore, in our experiments, all methods are evaluated fairly by performing attacks **at the early stage** of model training.
> > > > >
> > > > > However, it is important to note that under our current model design, the GIT generator can **reconstruct an arbitrary number of data instances**, whereas optimization-based and generative model–based methods require iterative optimization—meaning that in any training phase, each image must be reconstructed through an **independent optimization process**, resulting in high computational cost. In contrast, our approach trains an input–gradient mapping model, enabling **scalable data reconstruction**.
> > > > > ## Question 11:
> > > > > Given that the proposed technique and R-GAP are based on similar recursive formulae, the authors should compare their proposed methodology.
> > > > > ## Answer to Question 11:
> > > > > Please refer to the **answers to weakness 2.2**.
> > > > > ## Question 12:
> > > > > Can the authors discuss the implications of Equation 3.2?
> > > > > ## Answer to Question 12:
> > > > > Minor correction: we denote the outputs as $f_j^{\text{out}}$, and there is no $f_i^{\text{out}}$. Different networks have very different connectivity patterns, such as attention layers or residual connections, but all of these can be represented using a MIMO formulation. Therefore, we propose this **general form**, which is applicable to various types of networks. Once a module is selected (where a module can be a layer or a multi-head attention block), $f^{\text{in}}$ and $f^{\text{out}}$ are fixed. Equation 5 gives the reconstruction of a specific $f_i^{\text{in}}$ in the MIMO network of a layer, and thus depends on **all** corresponding $f_j^{\text{out}}$ and **one input connection** $f_i^{\text{in}}$. This finding is unsurprising or, one could say, rather expected.
> > > > >
> > > > > To clarify again: Our approach performs **layer-wise** or **module-wise reconstruction**, where each layer or module reconstruction corresponds to one MLP. The input to each MLP is determined by the theoretical formulas, thereby implicitly performing the reconstruction. Since the formulas in Section 3.1 apply to a single layer, and those in Section 3.2 apply to a single module, the GIT generator must be built iteratively.
> > > > >
> > > > > Because our method is **high-level architecture adaptive**, it is not possible to provide a single general formula. The construction of the generator is clearly described in Algorithm 1, where the iterative procedure explicitly defines the method.

---

> ### Comment · Reviewer_xnwp · 2025-11-24
>
> I thank the authors for their response. I, however, remain unconvinced by the arguments they presented and by the fact that they haven't started updating the paper with what they promise to eventually revise it with. I have included here my response to their rebuttal. If a question/weakness is unaddressed, it can be considered resolved.
>
> ## Weakness 1
>
> The comments the authors present must be accommodated in the paper.
>
> ## Weakness 2.2
>
> For this point, I would like to clarify that I am mostly referring to remaining concerns in other sections, namely:
>
> - The lack of an explicitly defined threat model.
> - The missing derivations for Eq.3 and 5.
> - The missing experimental details, such as image size, or batch size.
>
> These need to be included in the paper, without which conveying the contribution becomes challenging, and unclear.
>
> ## Weakness 2.4
>
> The authors do not acknowledge this, albeit minor concern, and have not worked on addressing it neither in their rebuttal, nor in their paper.

---

> ### Comment · Reviewer_xnwp · 2025-11-24
>
> ## Weaknesses 3.1-3.5
>
> My concerns regarding the threat model have not been addressed by the authors' response. While I understand the essence of their setting, which includes a single hacked client, I am still unconvinced by their motivations, the realism of the setting, and the underlying assumptions.
>
> In particular, from what I understand GIT operates on a "black-box setting", but assumes:
>
>  - The hacker has write-only access to the client's training data, so as to impute/replace the public data to the private one, and observe the gradients through the model.
>
>  - Access to the gradient reporting channel.
>
>  - Knowledge of the model's architecture, and structure.
>
>
> I have a couple of concerns with this notion.
>
> First, I was unable to find a section in **SoK: Gradient Inversion Attacks in Federated Learning** that describes the particular setting, despite the authors' claim that one exists. I would be grateful if the authors could point it out.
>
> Second, I do not believe the setting is realistic given the assumptions. The authors argue that we can assume an attack similar either to a standard backdoor or data poisoning attacks. For one, if we assume a standard backdoor, then it does not seem feasible that the attacker has only write-access, without being able to directly just observe the private data. On the other hand, the classical notion of data poisoning involves access for modifying training data, which realistically involves read access as well. Data poisoning attacks in FL involve **malicious clients**, which poison the model to negatively effect the global model, harming **other clients**, and has nothing to do with external tampering of a benign client's public data. Therefore, the authors must specify and cite the specific instance of data poisoning, in order to validate their assumption.
>
> Third, the attack assumes knowledge a strong adversary that has access to both the model architecture, the input to the FL training pipeline, the reported gradients, but for some reason has no access to the model weights. It seems that the adversary has significantly elevated access to the client's system, so the authors must explain how it is realistic for the client to have no access to model weights.
>
> Finally, I doubt that this setting can easily go undetected. It requires the usage of the client's compute for non-FL related processing, in order to get sufficient data to train the GIT model. I doubt that a realistic client would not notice having a substantial amount of additional data imputed, or significant additional compute use.
>
> In contrast, the honest-but-curious setting requires that a server be compliant, but still wants to recover the client's private data. This is undetectable, as the server does not leak any information, and is a risk that must be assumed in an FL setting.
>
> Here I list some actionable points from the discussion above:
>
> - Can the authors **clearly list and thoroughly describe each assumption**, and justify each one?
>
> - Can the authors point me to the section of **SoK: Gradient Inversion Attacks in Federated Learning**, which discusses data injection, and cites the relevant paragraph. The only reference I was able to find with a similar notion, was a defence proposal, where the client can mix public data with the private data to increase the number of local update rounds, before sharing the gradients.
>
> - Given the most standard setting of gradient leakage attacks is simply the requirement for existing gradients, the authors' assumptions for how they can obtain other clients' information are strictly stronger.
>
> - "As illustrated in Figure 1, we adopt a similar premise to DLG Zhu et al. (2019): the attacker hacks the channel to inject data to one client, which shares the gradient with the server and other clients" -> this statement in L68-69 of the paper remains untrue, as the authors claim that their setting differs to white-box settings, such as DLG, but do not address the included statement.
>
> - In their response to **Weakness 3.1**, the authors refer to membership inference attacks as an example for a black-box setting. While there do exist MIAs that can discern whether a sample gradient has come from a given model, this is nevertheless a binary classification task, which is not trivially relevant to gradient leakage attacks. Can the authors elaborate further on this point?
>
> ## Weakness 4
>
> The authors are free to use the extra page, given for post-review modifications, to address the issues, or at least acknowledge/refute the concerns in their rebuttal. For now, this point is left basically unaddressed.

---

> ### Comment · Reviewer_xnwp · 2025-11-24
>
> ## Weakness 5 + Questions 7-8
>
> I agree with the authors' response that the system is indeed not underdetermined, as I was initially left with the impression that Exact-GIT only tried to perform the exact recovery through a single system of equations. However, I remain unconvinced by the remaining points the authors convey here. They claim that:
>
> - "Exact-GIT explicitly estimates the unknown ground-truth weights": While the original weights present a global minimum for the optimization problem, given the number of parameters, compared to the number of training points, it is unclear how the training dynamics allow for actually recovering a similar model to the leaked one.
>
> - "The motivation for Exact-GIT is solely to demonstrate the feasibility of performing explicit weight estimation in a black-box scenario": While the authors report a convergence in the L2 distance between the leaked model and the generative model, it is difficult to contextualise its implications, given that 1) the scale and size of the original weights is not provided, making the L2 distance uninterpretable, and 2) it is unclear whether the "recovered" weights have any practical use.
>
> - "this comes with well-known numerical instability due to the use of pseudo-inverse operations" - This points remains vague and unsubstantiated. For example, R-GAP takes advantage of a similar recursive pattern and successfully applies the Moore-Penrose pseudoinverse to achieve near-exact reconstructions. Can the authors present quantitative evidence that has led them to this claim? Providing the reconstruction results, as mentioned in the next point, and isolating the pseudoinverse calculation as the critical failing point seem necessary to make this claim.
>
> - The authors never answered the question posed in **Question 7**, namely what the reconstruction metrics for Exact-GIT are. To claim any use for the proposed Exact-GIT, the authors must addressed the previous points I made in this response AND provide the **numerical results, similar to Table 2**.
>
> ## Weakness 6.1
>
> The batch size needs to be specified in the main text, in the main experiments, not simply in the caption of a table, a subsection after the main results. This makes interpreting the results more challenging, and left me guessing whether the batch size of 1 held for all settings.
>
> Furthermore, the original ImageNet resolution is undefined, as the images are of different sizes. From the supplementary material (specifically L502-506), it can be seen that the datasets have been preprocessed to fit a $224 \times 224$ size, which should be clearly clarified in the paper. While this size is relatively standard, it is not a consensus that every work uses this size. As the results are heavily dependent on this preprocessing step, the authors must include this in their experimental details.
>
> ## Weakness 6.3
>
> Equation 3 is specifically used for a MIMO architecture, which the ViT does not fall under, while it is also unclear how Eq. 5 would relate to a transformer-based architecture practically. This renders the theoretical inspiration unfounded in this case. Given that the authors observe a similar improvement for ViT as for the other networks, it begs the question of whether the neural network choice, as inspired by Eqs. 3,5, improves the performance, in contrast to an arbitrary architecture.
>
> ## Weakness 6.5
>
> I suppose that the authors are referring to Table 5. In any case, the FER-JAFFE comparison still operates on the same task, namely Facial Expression Recognition, which have similar classification labels, as well as having FER containing images, which are close to JAFFE in distribution regarding composition, race, and gender.
>
> ## Question 7
>
> Repeating the point from above, can you explicitly report the reconstruction results of Exact-GIT, similar to those in Table 2?
>
> ## Question 8
>
> Can the authors include the code they used for running the experiment in Figure 3?
>
> ## Question 9
>
> Can the authors elaborate more here? How does the LeNet architecture, and the CIFAR10 dataset compare to the one presented in IG, so that there are any differences between the scores? Can you include this ablation study you mention regarding the different kernel sizes and step sizes?
>
> ## Question 10
>
> The point you are trying to make, in particular that "the GIT generator can reconstruct an arbitrary number of data instances", depends on how GIT's performance degrades over the FL training without retraining the generator. More precisely, after 10 training steps, the "leaked model"'s weights and the corresponding produces gradients have changed, meaning they do not fit the distribution that the GIT generator expects anymore. To confirm the scalability of their approach, the authors should evaluate how the GIT generator performs over the FL training period. Otherwise, the trained model might only be useful for a couple of training steps before needing finetuning/retraining.

---

> ### Comment · Reviewer_xnwp · 2025-11-24
>
> ## Additional questions
>
> - The authors have reported results for larger batch sizes in their response to reviewer **4vGz**. However, in their code and paper, they describe being able to recover the output labels through the last-layer bias. Furthermore, in their response to reviewer **4vGz**, they mention as a weakness for "See through gradients: Image batch recovery via gradinversion", that it requires non-overlapping labels within a batch. How do you handle this for the larger batch sizes, where the information in the last-layer bias is mixed, without assuming non-overlapping labels?
>
> - The authors claim in their response to W6.2 that they used a non-pretrained GIAS for their comparisons. This doesn't seem like a fair comparison, given that they allow GIT to be trained on the public portion of the dataset. How would the results change if GIAS is also trained on the public dataset, but has the private data held out?
>
> ## Summary
>
> Given the authors' unwillingness to include any quantitative evidence and the fact that they have not yet updated neither the paper nor their supplementary material leaves me deeply unconvinced by their arguments in the rebuttal.

---

### Official Review · Reviewer_4vGz · 2025-10-29

**Soundness:** 3
**Presentation:** 3
**Contribution:** 2
**Rating:** 4
**Confidence:** 3

**Summary:**

This submission introduces Gradient Inversion Transcript (GIT), a model-based attack to reconstruct training data from leaked gradients in federated learning. The main idea is to construct an auxiliary reconstruction model whose architecture is adaptively designed to approximate the inverse of the target model's back-propagation process. The authors present two variants: "Exact-GIT," a theoretically-grounded but computationally intensive version, and "Coarse-GIT," a more practical and efficient version that uses shallow MLPs to approximate the inversion at each layer or module. The experimental results demonstrates that GIT can be used for direct and efficient data reconstruction or as a powerful generator of priors to significantly boost the performance of traditional optimization-based attacks.

**Strengths:**

- Direct but Effective idea: Mirroring the inverse of the target model's gradient flow is interesting. The theoretical derivation in Section 3, which forms the basis of the method, is sound and clearly explained.
- Strong Performance: The empirical evaluation is comprehensive and demonstrates the effectiveness of GIT. 1) Direct Inference: As shown in Table 2, GIT consistently outperforms both optimization-based methods (DLG, IG) and other model-based methods (LTI) in direct reconstruction across two datasets (CIFAR-10, ImageNet), multiple metrics (MSE, PSNR, SSIM), and different architectures (LeNet, ResNet, ViT); 2) As a Prior: When combined with an optimization method (GIT+IG), the approach achieves state-of-the-art results, significantly outperforming other hybrid methods like GIAS+IG. This shows that GIT generates higher-quality initial estimates.

**Weaknesses:**

- Maybe outdated baseline: The latest baseline mentioned in this submission was presented in 2023. Is there any new state-of-the-art methods?
- Practical Complexity: The reconstruction model's architecture must be custom-built for each target model. Is the architecture design of the Coarse-GIT affect the performance?
- Computational Cost: The training efficiency of GIT is worse than DLG and IG.

**Questions:**

- The GIT+IG significantly outperforms GIAS+IG. GIAS relies on a pre-trained generative model, whereas GIT does not. What are the reasons that GIT provides better priors? Is it primarily because its architecture is tailored to the inversion task, whereas a standard GAN architecture is too generic?
- The effectiveness of gradient attacks is affected by the batch size, since gradients are averaged over more samples. Your experiments seem to focus on small batch sizes. Could you provide empirical results or discuss how the performance of GIT degrades as the batch size increases?

---

> ### Author Response · Authors · 2025-11-21
>
> ## Weaknesses 1:
> Maybe outdated baseline: The latest baseline mentioned in this submission was presented in 2023. Is there any new state-of-the-art methods?
> ## Response to Weakness 1:
> The comparison results of the latest generative-based methods are shown below:
>
> | Dataset  | Leaked Model | Method | MSE ↓  | PSNR ↑ | LPIPS ↓ | SSIM ↑ |
> |----------|--------------|--------|--------|--------|----------|---------|
> | ImageNet | ResNet       | IG     | 0.161  | 9.17   | 0.8802   | 0.1283  |
> | ImageNet | ResNet       | GIAS   | 0.037  | 14.32  | 0.8218   | 0.3765  |
> | ImageNet | ResNet       | GIFD   | 0.027  | 15.90  | 0.7125   | 0.4479  |
> | ImageNet | ResNet       | LTI    | 0.029  | 15.38  | 0.7434   | 0.4129  |
> | ImageNet | ResNet       | GIT    | 0.021  | 16.78  | 0.6995   | 0.4758  |
> > Caption: Comparison results of generative-based methods (GIAS & GIFD) and gradient-input mapping methods (LTI & GIT) on Imagenet.
> ## Weaknesses 2:
> Practical Complexity: The reconstruction model's architecture must be custom-built for each target model. Is the architecture design of the Coarse-GIT affect the performance?
> ## Response to Weakness 2:
> Coarse-GIT differs from directly using a single MLP to approximate the entire inverse backpropagation process—that is, mapping gradients back to the input. Instead, we train an MLP for each **module** of the target model: a layer in LeNet, a layer in ResNet, or a multi-head attention block in ViT. Our method also adapts to high-level architectural structures: the connections between these MLPs follow the original model topology, including components such as residual blocks. This **high-level** architecture adaptation leads to better performance compared with approaches that ignore architectural structure.
> ## Weaknesses 3:
> Computational Cost: The training efficiency of GIT is worse than DLG and IG.
> ## Response to Weakness 3:
> All optimization-based and generative model-based attacks incur **computational costs that scale with the reconstruction workload**, whereas GIT requires **only a single upfront training and its training cost remains constant** regardless of how many reconstructions are performed.
>
> Specifically, for optimization-based and generative model-based attacks, let $E_{gm}$ denote the number of reconstruction iterations for each data batch, Reconstructing $N$ batches of input instances then requires $2N E_{gm}$ forward or backward passes (this refers to inference time, as these methods do not require training an auxiliary model). \
> In contrast, the computational complexity of our proposed GIT mainly depends on the number of training iterations. If the GIT generator is trained for $E_{g}$ epochs with $B$ data batches per epoch, the training cost amounts to $2 B E_{g}$ forward or backward passes (training time). During inference, only $N$ forward passes are needed to reconstruct $N$ batches of input instances, which is negligible compared to the training cost.
> The inference process only require $N$ forward passes to reconstruct $N$ batches of input instances (inference time), which is nearly negligible compared with training cost.
> Therefore, the total computational cost for generative approaches can be expressed as:
>
> $$(2B E_{g} + N) \ \text{forward or backward passes}$$
>
> It is worth noting that $E_{g} \ll E_{gm}$ and $N \simeq B$ in practice, which leads to an exaggerated computational cost for optimization-based and generative model-based attacks when reconstructing multiple batches.
>
> Besides the theoretical analysis, **empirically**, as shown in **Table 2** and **Table 3**, we found that GIT's inference time is lower than that of other methods, demonstrating its scalable efficiency.

---

> > ### Author Response · Authors · 2025-11-21
> >
> > ## Question 1:
> > The GIT+IG significantly outperforms GIAS+IG. GIAS relies on a pre-trained generative model, whereas GIT does not. What are the reasons that GIT provides better priors?
> > ## Answer to Question 1:
> > Generative model–based methods, including diffusion-based and GAN-based approaches, effectively map a latent space into high resolution images, thereby reducing the difficulty of reconstruction, which makes a direct comparison with our proposed method unfair. For this reason, in our experiments, the GAN baseline is used **without pretraining** (as stated in Lines 374–375).
> >
> > It is also important to note that generative model–based methods are similar in nature to DLG in that they require **iterative optimization**, meaning each image must be reconstructed through a separate optimization process, which incurs high computational cost. In contrast, our method trains an input–gradient mapping model, which needs only one forward propagation for gradient inversion, enabling **scalable data reconstruction**.
> > ## Question 2:
> > Could you provide empirical results or discuss how the performance of GIT degrades as the batch size increases?
> > ## Answer to Question 2:
> > Results for larger batch sizes are shown below. More experiments will be added into the revised PDF. Some methods can handle very large batch sizes, such as **Spear** or **See through gradients: Image batch recovery via gradinversion**. However, these approaches require additional information—e.g., ReLU activations and greedy search algorithms for Spear, or no overlapping labels within a batch (as in See Through Gradients)—which often requires additional knowledge in practice.
> > | Model | Method | Metrics | BS=2 | BS=4 |
> > | :---- | :---- | :---- | :---- | :---- |
> > | LeNet | LTI | MSE | 0.028 | 0.037 |
> > |  |  | PSNR | 15.53 | 14.32 |
> > |  | GIT | MSE | 0.016 | 0.018 |
> > |  |  | PSNR | 17.96 | 17.45 |
> > | ResNet | LTI | MSE | 0.061 | 0.063 |
> > |  |  | PSNR | 12.15 | 12.01 |
> > |  | GIT | MSE | 0.052 | 0.052 |
> > |  |  | PSNR | 12.84 | 12.79 |
> > > caption: Comparison results of LTI and GIT with different batch sizes

---

### Official Review · Reviewer_jgr4 · 2025-10-30

**Soundness:** 3
**Presentation:** 3
**Contribution:** 3
**Rating:** 6
**Confidence:** 4

**Summary:**

The paper proposes Gradient Inversion Transcript (GIT), a model-based method for reconstructing private training data from leaked gradients. GIT builds a reconstruction model that mirrors the inverse of back-propagation and adapts its architecture to the attacked model. Trained offline on public input–gradient pairs, GIT reconstructs data efficiently without requiring model parameters or labels. Two variants are introduced—Exact-GIT (theoretical inversion) and Coarse-GIT (approximate, efficient). Experiments on CIFAR-10, ImageNet, and facial datasets with LeNet, ResNet, and ViT show that GIT achieves superior performance and strong robustness to noisy gradients and distribution shifts. Moreover, using GIT outputs as priors for iterative optimization further improves accuracy and convergence.

**Strengths:**

-The proposed solution is conceptually sound and appears novel.

-The evaluation setting is extensive. The authors conduct experiments across multiple architectures and cover diverse datasets. The paper also goes beyond standard settings by testing under challenging conditions.

-The proposed method shows consistently superior results across datasets and architectures, outperforming both optimization- and model-based baselines.

**Weaknesses:**

-The comparison baselines are relatively outdated, mostly limited to earlier methods such as DLG, IG, and LTI. The paper lacks comparisons with more recent or stronger baselines, making it difficult to assess whether the proposed method remains competitive.

-The paper assumes the attacker can both inject data into a client and obtain structured per-layer gradients, which requires high-level access and control. This dual assumption may be overly strong or unrealistic in many real-world FL deployments.

**Questions:**

-The problem formulation is not entirely clear. Does the proposed attack allow the adversary to reconstruct data from all clients in the federated system, or only from the compromised one? If it can access data from other clients, please clarify the underlying assumption or mechanism that enables this cross-client reconstruction.

-The paper claims that GIT is adaptive to the leaked model’s architecture, but this seems to apply only to Exact-GIT. Is Coarse-GIT also adaptive in any way, or does it merely use a fixed MLP structure? If the latter, it appears to lose the key advantage of architectural adaptivity.

-The experimental section seems to mix results from Exact-GIT and Coarse-GIT, which makes it unclear which variant contributes to the reported performance.

---

> ### Author Response · Authors · 2025-11-21
>
> ## Weaknesses 1:
> The comparison baselines are relatively outdated.
> ## Response to Weakness 1:
> We have included GIAS [1] in **Table 3** as the representative baseline among the generative model-based methods. For the other SOTA methods, we provide the supplementary results below:
>
> | Dataset  | Leaked Model | Method | MSE ↓  | PSNR ↑ | LPIPS ↓ | SSIM ↑ |
> |----------|--------------|--------|--------|--------|----------|---------|
> | ImageNet | ResNet       | IG     | 0.161  | 9.17   | 0.8802   | 0.1283  |
> | ImageNet | ResNet       | GIAS   | 0.037  | 14.32  | 0.8218   | 0.3765  |
> | ImageNet | ResNet       | GIFD   | 0.027  | 15.90  | 0.7125   | 0.4479  |
> | ImageNet | ResNet       | LTI    | 0.029  | 15.38  | 0.7434   | 0.4129  |
> | ImageNet | ResNet       | GIT    | 0.021  | 16.78  | 0.6995   | 0.4758  |
> > Caption: Comparison results of generative-based methods (GIAS & GIFD) and gradient-input mapping methods (LTI & GIT) on Imagenet.
> ## Weaknesses 2:
> The paper assumes the attacker can both inject data into a client and obtain structured per-layer gradients, which requires high-level access and control.
> ## Response to Weakness 2:
> As summarized from **Table 1**, existing optimization-based and generative model–based attacks typically rely on stronger assumptions: Our proposed GIT assumes access only to gradient queries (a black-box setting), whereas methods such as GIAS, DGGI, GIFD and DLG [1, 2, 3, 4] also assume gradient queries **plus additional access to data labels**. R-Gap [5], in contrast, requires full **access to model parameters**, corresponding to a white-box setting.
> Compared with these approaches, GIT relies on strictly weaker assumptions than those requiring parameter access and is comparable to—or even milder than—the assumptions used by prior methods. Thus, GIT operates under **the least amount of prior knowledge** among these reconstruction approaches.
> ## Question 1
> Does the proposed attack allow the adversary to reconstruct data from all clients in the federated system, or only from the compromised one? If it can access data from other clients, please clarify the underlying assumption or mechanism that enables this cross-client reconstruction.
> ## Answer to Question 1:
> The proposed attack enables an adversary to reconstruct data from **all clients** in the federated system, as long as the compromised client can observe gradients originating from the target clients.
>
> The key difference between the hacked client and other clients is the access of gradient query. When reconstructing data from others, the attack suffers from **parameter discrepancy** and **data domain shifts**: in federated learning, each client maintains a slightly diverged local version of the shared global model, shaped by the client-specific data domain and local training process. This mismatch makes reconstruction from non-compromised clients more challenging. In addition, the gradients from other clients are obtained through either **decentralized gradient sharing** or **centralized gradient broadcasting**. Specifically, in centralized federated learning, an adversary controlling one client can receive other clients’ gradients through the server’s broadcast. In decentralized federated learning, an adversary acting as a regular participating client can similarly observe gradients shared by neighboring clients.
> ## Questions 2:
> The paper claims that GIT is adaptive to the leaked model’s architecture, but this seems to apply only to Exact-GIT. Is Coarse-GIT also adaptive in any way, or does it merely use a fixed MLP structure? If the latter, it appears to lose the key advantage of architectural adaptivity.
> ## Answer to Question 2:
> Coarse-GIT differs from directly using a single MLP to directly approximate the entire inverse backpropagation process—that is, mapping gradients to the input. Instead, we train an MLP for each **module** of the target model: a layer in LeNet, a layer in ResNet, or a multi-head attention block in ViT. Coarse-GIT adapts to the **high-level architecture** of the targeted model: The connections between these MLPs follow the original model’s high-level structure.
> ## Questions 3:
> The experimental section seems to mix results from Exact-GIT and Coarse-GIT.
> ## Answer to Question 3:
> To clarify, **only Figure 3 in Appendix F.2 reports results for Exact-GIT**. All experimental results in the main paper correspond to Coarse-GIT, as stated in Line 332. We adopt Coarse-GIT because, unlike Exact-GIT which requires multiple pseudo-inverse operations, it is more numerically stable and thus more suitable for large-scale experiments.

---

> > ### Author Response · Authors · 2025-11-21
> >
> > ## Reference
> > [1] Jinwoo Jeon, Kangwook Lee, Sewoong Oh, Jungseul Ok, et al. Gradient inversion with generative image prior. Advances in neural information processing systems, 34:29898–29908, 2021.
> >
> > [2] Hao Fang, Bin Chen, Xuan Wang, Zhi Wang, and Shu-Tao Xia. Gifd: A generative gradient inversion method with feature domain optimization. In Proceedings of the IEEE/CVF International Conference on Computer Vision, pages 4967–4976, 2023.
> >
> > [3] Liwen Wu, Zhizhi Liu, Bin Pu, Kang Wei, Hangcheng Cao, and Shaowen Yao. Dggi: Deep generative gradient inversion with diffusion model. Information Fusion, 113:102620, 2025.
> >
> > [4] Ligeng Zhu, Zhijian Liu, and Song Han. Deep leakage from gradients. In H. Wallach, H. Larochelle, A. Beygelzimer, F. d'Alché-Buc, E. Fox, and R. Garnett (eds.), Advances in Neural Information Processing Systems, volume 32. Curran Associates, Inc., 2019.
> >
> > [5] Junyi Zhu and Matthew Blaschko. R-gap: Recursive gradient attack on privacy. arXiv preprint arXiv:2010.07733, 2020.

---

### Official Review · Reviewer_zriV · 2025-11-02

**Soundness:** 2
**Presentation:** 2
**Contribution:** 3
**Rating:** 4
**Confidence:** 3

**Summary:**

The paper "Gradient Inversion Transcript (GIT)" presents a new framework for reconstructing training data from leaked gradients in distributed or federated learning settings. GIT models the mathematical process of back-propagation itself and learns a neural mapping that approximates its inverse, this mathematical derivation reveals that gradients encode sufficient information to estimate layer inputs through tensor operations and pseudo-inverses. GIT can reconstruct private input data directly from leaked gradients in one forward pass, achieving real-time inversion, outperforming prior state-of-the-art inversion attacks in reconstruction accuracy, robustness to noise, and computational efficiency.

**Strengths:**

Theoretically Grounded and Interpretable. GIT is derived from the back propagation equations, not designed heuristically. This gives it a strong theoretical foundation and interpretability
Architecture-Adaptive Modular Design. GIT decomposes the target model into multi-input multi-output (MIMO) modules. Each module learns to invert its corresponding back-propagation block, allowing GIT to adapt naturally to different model architectures
Cross-Client attack Capability. GIT can reconstruct inputs of other clients using shared or aggregated gradients by leveraging its learned understanding of how gradients encode input activations, achieving "one2many" inversion capability

**Weaknesses:**

Pretraining depends on public data. GIT requires public data to train the inversion model offline. If the public data are very different from the target domain, he reconstruction quality may degrade substantially.
Whether it is effective in the actual scene? The attack assumes knowledge of the model architecture and access to shared gradients. This setting may not adapt to fully secured or asynchrony.

**Questions:**

See weakness

---

> ### Author Response · Authors · 2025-11-21
>
> ## Weaknesses 1:
> The attack assumes knowledge of the model architecture and access to shared gradients. This setting may not adapt to fully secured or asynchrony.
> ## Response to Weakness 1:
> For empirical results under more challenging scenarios,  including the domain shift cases, please refer to **Section 5.3**.
>
> For an adversary who knows the task objective, the model architecture is typically easy to obtain. In our setting, the adversary possesses **the least amount of knowledge** compared with generative model–based methods and optimization-based methods, as summarized in **Table 1**. Existing optimization-based and generative model–based attacks typically rely on stronger assumptions. Our proposed GIT assumes access only to gradient queries (a black-box setting), whereas methods such as GIAS, DGGI, GIFD and DLG [1, 2, 3, 4] also assume gradient queries **plus additional access to data labels**. R-Gap [5], in contrast, requires full **access to model parameters**, corresponding to a white-box setting.
> Compared with these approaches, GIT relies on strictly weaker assumptions than those requiring parameter access and is comparable to—or even milder than—the assumptions used by prior methods. Thus, GIT operates under **the least amount of prior knowledge** among these reconstruction approaches.
>
> ## Reference
> [1] Jinwoo Jeon, Kangwook Lee, Sewoong Oh, Jungseul Ok, et al. Gradient inversion with generative image prior. Advances in neural information processing systems, 34:29898–29908, 2021.
>
> [2] Hao Fang, Bin Chen, Xuan Wang, Zhi Wang, and Shu-Tao Xia. Gifd: A generative gradient inversion method with feature domain optimization. In Proceedings of the IEEE/CVF International Conference on Computer Vision, pages 4967–4976, 2023.
>
> [3] Liwen Wu, Zhizhi Liu, Bin Pu, Kang Wei, Hangcheng Cao, and Shaowen Yao. Dggi: Deep generative gradient inversion with diffusion model. Information Fusion, 113:102620, 2025.
>
> [4] Ligeng Zhu, Zhijian Liu, and Song Han. Deep leakage from gradients. In H. Wallach, H. Larochelle, A. Beygelzimer, F. d'Alché-Buc, E. Fox, and R. Garnett (eds.), Advances in Neural Information Processing Systems, volume 32. Curran Associates, Inc., 2019.
>
> [5] Junyi Zhu and Matthew Blaschko. R-gap: Recursive gradient attack on privacy. arXiv preprint arXiv:2010.07733, 2020.

---

### Official Review · Reviewer_QwL7 · 2025-11-03

**Soundness:** 3
**Presentation:** 3
**Contribution:** 3
**Rating:** 6
**Confidence:** 3

**Summary:**

This paper studies the gradient leakage problem in the federated learning setting. The problem is challenging to reconstruct training images from leaked gradients. This is achieved by an auxiliary model, that GIT generator is trained with image/gradient pairs.  In inference, this GIT generator is applied to the leaked gradients produced from clients' data. Finally, reconstructed images can be obtained. To be honest, we did not check the correctness of all the mathematical formulations in this paper, but the experiments show the better attack performance. In the appendix, visualization results are also shown. In order to show the effectiveness of this method, CIFAR-10 and ImageNet are applied with different network architecture including LeNet, ResNet, and ViT.

**Strengths:**

1. The overall idea is clear and interesting. An auxiliary model GIT generator is trained to reconstruct images. The results show that the proposed method can at least recover some shape and color information from leaked gradients, even though many details are missing.

2. This paper is well organized and provides mathematical formulations. Also, reconstruction under different scenarios is discussed, including inaccurate gradients, distribution shift.

3. Dynamically selection of architecture of the threat model is interesting and important for achieving better performance.

**Weaknesses:**

1. The discussion on the reasons of focusing on input-gradient mapping-based methods is not sufficient. Why does this paper focus on input-gradient mapping-based approach? The reconstructed images from GIT can be applied by diffusion models to produce reasonable image details. From the visualization results in appendices, the reconstructed images lack of enough details.

2. The motivation is to estimate clients' training data in federated learning settings. The discussion in federated learning settings is not enough. It seems the proposed method can be also used to attach centralized training.

**Questions:**

1. The distribution of gradients is affected by many factors, including the stage of model training, the dataset distribution, and network architectures. How to choose a GIT generator for different stages of federated learning, for example early training with large learning rate or finetuning with smaller learning rate?

---

> ### Author Response · Authors · 2025-11-21
>
> ## Weaknesses 1:
> Why not using generative model?
> ## Response to Weakness 1:
> Generative model–based methods, including diffusion-based and GAN-based approaches [1, 2, 3, 4], effectively map a latent space to high-resolution images, thereby reducing the difficulty of reconstruction. However, they are similar in nature to DLG [5] in that they require iterative optimization (the number of iterations required depends on both the attack method and the image resolution. For example, on CIFAR-10 with an SGD-based optimization attack, it typically takes around 10 million iterations per image to converge), which means each image must be reconstructed through an independent optimization process, resulting in high computational cost. In contrast, our method trains an input–gradient mapping model. Once trained, the GIT model only needs one-time inference to do gradient inversion for one image, enabling **scalable data reconstruction**.
>
> Although our method is generic and can be combined with these generative models in the same way as GIT+DLG, we do not include the diffusion-model as baselines due to the time-consuming diffusion process.
> For the GANs, we include GIAS in **Table 3**, which employs a GAN *without pretraining* for fairness.
> ## Weaknesses 2:
> Whether the applicable settings are limited to federated learning (FL)?
> ## Response to Weakness 2:
> Yes, GIT can also be applied to centralized learning. Our method can be used in settings where both the model architecture and the gradients are exposed to the attacker. We emphasize the FL scenario because gradient communication makes gradient leakage far more likely, so FL is more susceptible to such attacks.
> ## Question 1:
> How to choose a GIT generator for different stages of federated learning?
> ## Answer to Question 1:
> For different stages of the federated learning process, we need to train a corresponding GIT generator. This is because GIT performs the inversion of backpropagation, which is inherently affected by the leaked model parameters. Existing work (CENSOR: Defense Against Gradient Inversion via Orthogonal Subspace Bayesian Sampling) has shown that as training progresses, all gradient inversion attacks become increasingly difficult, since the gradients contain less information. Therefore, in our experiments, all methods are evaluated fairly by performing attacks at the early stage of model training. [1, 2, 3, 4, 5] also report their results based on the early-stage model.
> ## References
> [1] Hongxu Yin, Arun Mallya, Arash Vahdat, Jose M Alvarez, Jan Kautz, and Pavlo Molchanov. See through gradients: Image batch recovery via gradinversion. In Proceedings of the IEEE/CVF conference on computer vision and pattern recognition, pages 16337–16346, 2021.
>
> [2] Jinwoo Jeon, Kangwook Lee, Sewoong Oh, Jungseul Ok, et al. Gradient inversion with generative image prior. Advances in neural information processing systems, 34:29898–29908, 2021.
>
> [3] Hao Fang, Bin Chen, Xuan Wang, Zhi Wang, and Shu-Tao Xia. Gifd: A generative gradient inversion method with feature domain optimization. In Proceedings of the IEEE/CVF International Conference on Computer Vision, pages 4967–4976, 2023.
>
> [4] Liwen Wu, Zhizhi Liu, Bin Pu, Kang Wei, Hangcheng Cao, and Shaowen Yao. Dggi: Deep generative gradient inversion with diffusion model. Information Fusion, 113:102620, 2025.
>
> [5] Ligeng Zhu, Zhijian Liu, and Song Han. Deep leakage from gradients. In H. Wallach, H. Larochelle, A. Beygelzimer, F. d'Alché-Buc, E. Fox, and R. Garnett (eds.), Advances in Neural Information Processing Systems, volume 32. Curran Associates, Inc., 2019.

---

### Meta-Review · Area_Chair_jHSj · 2026-01-04

**Summary:**

Strengths mentioned by the reviewers:
- Dynamically selection of architecture of the threat model is interesting and important for achieving better performance.
- Theoretically Grounded and Interpretable.
- The proposed solution is conceptually sound and appears novel.
- The evaluation setting is extensive. The authors conduct experiments across multiple architectures and cover diverse datasets. The paper also goes beyond standard settings by testing under challenging conditions.
- Direct but Effective Idea. Idea to construct an inversion process by modeling the recursive relationship is creative and potentially valuable improvement over existing work. The mathematics is sound.
- The authors evaluating GIT not only as a standalone attack but in combination with existing optimization methods.
- Strong Performance: It seems that GIT is substantially more effective compared to existing attacks.

Weaknesses mentioned by the reviewers:
- Why does this paper focus on input-gradient mapping-based approach? The reconstructed images from GIT can be applied by diffusion models to produce reasonable image details. From the visualization results in appendices, the reconstructed images lack of enough details. **Partially addressed.**
- The motivation is to estimate clients' training data in federated learning settings. The discussion in federated learning settings is not enough. It seems the proposed method can be also used to attach centralized training. **Addressed.**
- Pretraining depends on public data. What if there is a domain shift? **Not addressed.**
- The attack assumes knowledge of the model architecture and access to shared gradients. This setting may not adapt to fully secured or asynchrony. **Addressed.**
- The paper assumes the attacker can both inject data into a client and obtain structured per-layer gradients, which requires high-level access and control. This dual assumption may be overly strong or unrealistic in many real-world FL deployments. The problem setting is unorthodox. **Partially addressed.**
- The baseline may be outdated. **Addressed.**
- Practical Complexity: The reconstruction model's architecture must be custom-built for each target model. Is the architecture design of the Coarse-GIT affect the performance? **Addressed.**
- Computational Cost: The training efficiency of GIT is worse than DLG and IG. **Addressed.**
- Details in Section 3 are unclear and contain factually wrong math. **Addressed**
- Writing and clarity can be improved. **Addressed**
- The assumptions are stronger than stated. **Partially addressed.**
- The presentation of the related work is confusing and at times misleading. **Not addressed.**

Questions raised by the reviewers:
- The distribution of gradients is affected by many factors, including the stage of model training, the dataset distribution, and network architectures. How to choose a GIT generator for different stages of federated learning, for example early training with large learning rate or finetuning with smaller learning rate? **Partially addressed.**
- The problem formulation is not entirely clear. Does the proposed attack allow the adversary to reconstruct data from all clients in the federated system, or only from the compromised one? If it can access data from other clients, please clarify the underlying assumption or mechanism that enables this cross-client reconstruction. **Partially addressed.**
- The paper claims that GIT is adaptive to the leaked model’s architecture, but this seems to apply only to Exact-GIT. Is Coarse-GIT also adaptive in any way, or does it merely use a fixed MLP structure? If the latter, it appears to lose the key advantage of architectural adaptivity. **Addressed.**
- The experimental section seems to mix results from Exact-GIT and Coarse-GIT, which makes it unclear which variant contributes to the reported performance. **Addressed.**
- The GIT+IG significantly outperforms GIAS+IG. GIAS relies on a pre-trained generative model, whereas GIT does not. What are the reasons that GIT provides better priors? Is it primarily because its architecture is tailored to the inversion task, whereas a standard GAN architecture is too generic? **Partially addressed.**
- The effectiveness of gradient attacks is affected by the batch size, since gradients are averaged over more samples. Your experiments seem to focus on small batch sizes. Could you provide empirical results or discuss how the performance of GIT degrades as the batch size increases? **Not addressed.** While the authors provide BS=2 and BS=4 it remains unclear to what extend the method scales to larger batch sizes and when it fails.
- What is the setting, in which GIT operates? The authors should clarify all assumptions for the attacker and system explicitly and clearly. **Addressed.**
- The authors must provide further experimental details. **Partially Addressed.**
- What are the reconstruction results for Exact-GIT, and how is the weight convergence relevant to them? **Partially Addressed.**
- The supplementary material provides an implementation for the MLP and ResNet variation of what seems like Coarse-GIT, there seem to be no details on the setup for Exact-GIT. Given that most of theoretical development is unused in Coarse-GIT, not having the option to investigate and replicate the Exact-GIT implementation is unfortunate. Can the authors provide additional clarification, and add the implementation to the supplementary material? **Partially Addressed.**
- The authors report lower PSNR for Geiping et al. [2] for CIFAR10 with the LeNet network than the original paper, even compared to batch sizes of 1, posing the evaluation methodology into question. Why is that the case? **Partially Addressed.**
- As training goes on, the model weights change, and hence the gradients of any input will change in comparison to the initial model state. How does a GIT, trained on the initial model state, perform over the FL training period? **Partially Addressed.**
- Can the authors compare their proposed methodology to that of R-GAP? Given that both techniques propose methods based on similar recursive formulae, the authors should make this clear. **Partially Addressed.**
- Given that Equation 3.2 does not depend on $f_i^\text{out}$, can the authors discuss the implications of this finding? This seems to imply that depending on how one picks  $f^\text{in}$ and $f^\text{out}$, it might result in a system of varying difficulty. **Partially Addressed.**

**Reviewer Concerns:**

See above.

**Reviewer Scores:**

- Reviewer QwL7: $6 \to 6$. Not all points where fully addressed.
- Reviewer zriV: $4 \to 4$. Some points remained unaddressed.
- Reviewer jgr4: $6 \to 6$. Not all points where fully addressed. The weaknesses raised by the reviewer where not insignificant.
- Reviewer 4vGz: $4 \to 4$. While most points where addressed, question 2 was not or at best partially addressed.
- Reviewer xnwp: $2 \to 2$. Many points remained only partially addressed. No quantitative evidence was provided.

---

### Decision · Program_Chairs · 2026-01-26

Reject